# Bacterial warfare is associated with virulence and antimicrobial resistance

Connor Sharp [1,2,3] & Kevin R. Foster [1,2,4]

Bacteria have evolved a diverse array of mechanisms to inhibit and kill competitors. However, why some bacteria carry such weapons while others do not remains poorly understood. Here we explore this question using the genomics of the bacteriocins of *E. coli* as a model system, which have large well-annotated bioinformatic resources. While bacteriocins occur widely, we find that carriage is particularly associated with pathogenic extra-intestinal (ExPEC) strains. These pathogens commonly carry large plasmids encoding bacteriocins alongside virulence factors and antimicrobial resistance mechanisms. Across all strains, we find many orphan immunity proteins, which protect against bacteriocins and suggest that these bacterial weapons are important in nature. We also present evidence that bacteriocin toxins readily move between strains via plasmid transfer and even between plasmids via transposons. Finally, we show that several *E. coli* bacteriocins are widely shared with the pathogen *Salmonella enterica*, further cementing the link to virulence. Our work suggests that the bacteriocins of *E. coli* are important antibacterial weapons for dangerous antimicrobial-resistant strains.

Bacteria often exist in dense and diverse communities where competition for nutrients or even just space to grow is intense[1–4]. In response to this competitive world, bacteria have evolved an arsenal of competition systems, also known as bacterial weapons, to inhibit each other[5–7]. These weapons can give bacteria an advantage over neighbouring cells and are important in determining the composition of bacterial communities[8,9]. The ability to invade or persist within a community is also critical for many bacterial pathogens, including in the human gut where the dense community of bacteria in the microbiome can limit pathogen growth[10–13]. While the potential benefits of carrying antibacterial weapons is clear, there is wide variation in both the number and types of weapons bacteria possess[7] and we have a limited understanding of why some bacteria are heavily armed, while others are not[14–16].

There is a need, therefore, to understand the role of bacterial weapons in the ecology of bacteria. Here we approach this challenge using large genomic databases that have recently become available for the model species *Escherichia coli*. In addition to the quality of the

genomic data for *E. coli*, the bacteriocins of *E. coli* are amongst the best-studied of all bacterial weapons and there is already a detailed understanding of many of their targets and molecular mechanisms[17]. But while bacteriocins were first described nearly 100 years ago[18], we still understand little about their roles in natural systems. Currently, there are a range of conflicting arguments that bacteriocins are important for bacterial competition[19–21], do not impact bacterial communities at all[16,22,23], or even exist solely as selfish genetic elements, maintaining themselves through a toxin/immunity system at the cost of the host[24,25]. In addition to its genomic resources, *E. coli* is an ideal model system for investigating the role of bacterial weapons in the ecology of bacteria because there is a large amount of variability in both weapon carriage and pathogenicity within the species, which allows one to look for associations between weapon use and ecology while avoiding potentially confounding differences among species.

There are two main classes of *E. coli* bacteriocins—colicins and microcins—which are protein or smaller peptide toxins respectively, that target closely related strains and species. These bacteriocins are

[1]Department of Biology, University of Oxford, Oxford, UK. [2]Department of Biochemistry, University of Oxford, Oxford, UK. [3]School of Biological Sciences, University of Reading, Reading, UK. [4]Sir William Dunn School of Pathology, University of Oxford, Oxford, UK. e-mail: c.sharp@reading.ac.uk; kevin.foster@path.ox.ac.uk

encoded either on plasmids or in the bacterial chromosome as a toxin and co-expressed immunity gene to prevent self-intoxication. Colicin-like proteins are also found in other Gammaproteobacteria species[26], including in the important pathogens *Salmonella enterica*[27] and *Klebsiella pneumoniae*[28]. To enter a target cell, colicins and microcins hijack outer membrane proteins[29], whose typical function is to import nutrients such as $B_{12}$, nucleosides and iron[17]. Detecting bacteriocins on plasmids has, until recently been a limitation, due to the quality of data available[30–32]. However, the massive increase in plasmid sequencing and improvements in assembly means one can now perform thorough analysis of plasmid-borne traits. In particular, here we are able to study thousands of genomes and plasmids from two large databases in order to analyse bacteriocins across the *E. coli* species.

We find that *E. coli* bacteriocins are abundant in pathogenic strains and linked to known virulence factors such as siderophores, adhesins and enterotoxins, as well as large arrays of multiple anti-microbial resistance (AMR) genes. Importantly, we also find plasmids containing orphan immunity proteins, which protect against bacteriocin attack without producing toxins, and support the importance of bacteriocins for bacterial competition (as opposed to solely acting as a plasmid addiction system). Finally, our data suggest that horizontal gene transfer of bacteriocins occurs on multiple levels: between species, within species and even from plasmid-to-plasmid via the action of transposons. We conclude that bacteriocins are important for bacterial ecology, competition and virulence.

## Results

### Bacteriocins are widespread throughout *E. coli*

We curated a dataset of 2601 *E. coli* strains with complete genomes (1 contig for the chromosome and 1 for each plasmid) and scanned these genomes for bacteriocins: colicin and microcins, either on plasmids or in the chromosome (Table 1). This analysis identified 608 strains (23.4% of our *E. coli* genomes) that encode at least one bacteriocin. Bacteriocins are mostly encoded on plasmids, with over three times more strains encoding plasmid bacteriocins than chromosomally encoded bacteriocins (18.5% vs 6.1% of all strains). Focusing on the plasmids, there are 571 bacteriocin plasmids in the data set which encode 730 bacteriocin genes (plasmids can encode multiple different colicins and microcins) spread across 466 strains. Bacteriocins are also very widely distributed across *E. coli* strains and are present in all *E. coli* phylogroups: 75% of phylogroup G plasmids carried bacteriocins (21/28), 46.3% of B2 (173/374), 34.4% of B1 (166/482), 24% of C (22/93), and 12% of A (121/990) (Other or un-typeable phylogroups: 105/634). In addition, bacteriocins are present in >90% (38/44) of all sequence types (STs) which had more than 10 strains in the dataset (Fig. 1B). Similarly, we find bacteriocins in 121/183 of the predicted O-antigen types in our dataset, and in 13/13 of all O-antigen types present in 40 or more strains.

Previous estimates of bacteriocin carriage rates in *E. coli* vary considerably from 15 to 60%. However, sampling and method of identification also vary between estimates with some estimates based exclusively on human faecal samples or specific *E. coli* pathotypes[33,34]. Our analysis of thousands of *E. coli* genomes and plasmids suggests that differences in bacteriocin abundance among phylogroups may explain the different estimates of bacteriocin carriage among studies. For example, the high prevalence of phylogroup A in our dataset, which has low bacteriocin carriage (12%) may help to explain why our estimate is relatively low as compared to studies focussed on other phylogroups and pathogenic *E. coli*[34–37]. Another possible factor for our work is that plasmid sequences may have been lost from certain samples, although our quality control methods that selects for complete genomes should mean this is relatively unlikely. Loss of plasmid DNA during DNA extraction is a possibility that we cannot exclude because sequencing quality control cannot mitigate against this.

Whatever the case, our estimates of bacteriocin carriage are likely to be conservative.

Some types of bacteriocin are much more prevalent than others in the data. Colicins M, Ia, Ib, E1 and Microcin V are the most abundant (Fig. 1A), which agrees with similar studies using cohorts of *E. coli* strains[33,38]. A key implication of this pattern is that pore-forming and peptidoglycan precursor degrading toxins, which both disrupt membrane integrity were more common than nucleases (DNases, tRNAses and rRNAses). Nuclease colicins, such as the E-type colicins, which are often used a model system for colicin translocation[39] and protein-protein interactions[40], are relatively rare in our database, representing <10% of the identified colicins.

Colicins and microcins are typically studied separately, and each have their own literatures[41,42]. However, our analyses suggest that they are functionally linked, with a strongly significant association between microcin and colicin carriage at the strain level (binaryPGLMM: $\beta = 2.07$, SE = 0.188, $p = 0.001$). A driver of this pattern is that most strains which encode a microcin on a plasmid also have a plasmid-borne colicin (63.5% of microcin carrying strains, Fisher exact test: OR 12.67, $p = <0.001$). Taken from the plasmid perspective, this manifests as 48.3% of microcin plasmids also encoding a colicin, e.g. a plasmid may encode genes for the microcin MccV and colicin ColM. These

**Table 1 | Bacteriocins differ in their targeted receptor and the cytotoxic domain they use to kill sensitive cells**

| Bacteriocin | Receptor | Cytotoxic Activity | Genetic Location |
|---|---|---|---|
| ColM | FhuA | Peptidoglycan precursor degrading | Plasmid |
| ColIa | Cir | Pore-formation | Plasmid |
| ColIb | Cir | Pore-formation | Plasmid |
| ColB | FepA | Pore-formation | Plasmid |
| ColE1 | BtuB | Pore-formation | Plasmid |
| ColS4 | OmpW | Pore-formation | Plasmid |
| ColN | OmpF | Pore-formation | Plasmid |
| ColK | Tsx | Pore-formation | Plasmid |
| Col5 | Tsx | Pore-formation | Plasmid |
| ColJs | - | Pore-formation | Plasmid |
| ColY | - | Pore-formation | Plasmid |
| CloDF13 | IutA | rRNase | Plasmid |
| ColD | FepA | rRNase | Plasmid |
| ColE3 | BtuB | rRNase | Plasmid |
| ColE6 | BtuB | rRNase | Plasmid |
| ColE2 | BtuB | DNase | Plasmid |
| ColE7 | BtuB | DNase | Plasmid |
| ColE8 | BtuB | DNase | Plasmid |
| MccV (Colicin V)[85] | Cir | IM depolarisation | Chromosome |
| MccB17 | OmpF | DNA Gyrase Inhibition | Plasmid |
| MccJ25 | FhuA[136] | RNA polymerase inhibition[137] | Plasmid |
| MccH47[138] | Cir, Fiu, FepA and IroN | ATP Synthase disruption | Chromosome |
| MccI47 | Cir, Fiu, FepA and IroN[139] | - | Chromosome |
| MccM | Cir, Fiu, FepA and IroN[139] | inhibition of oxidative phosphorylation[140] | Chromosome |

Bacteriocins can also either be plasmid encoded or chromosomal. '-' indicates an unknown receptor. Information is from[17], unless otherwise stated.

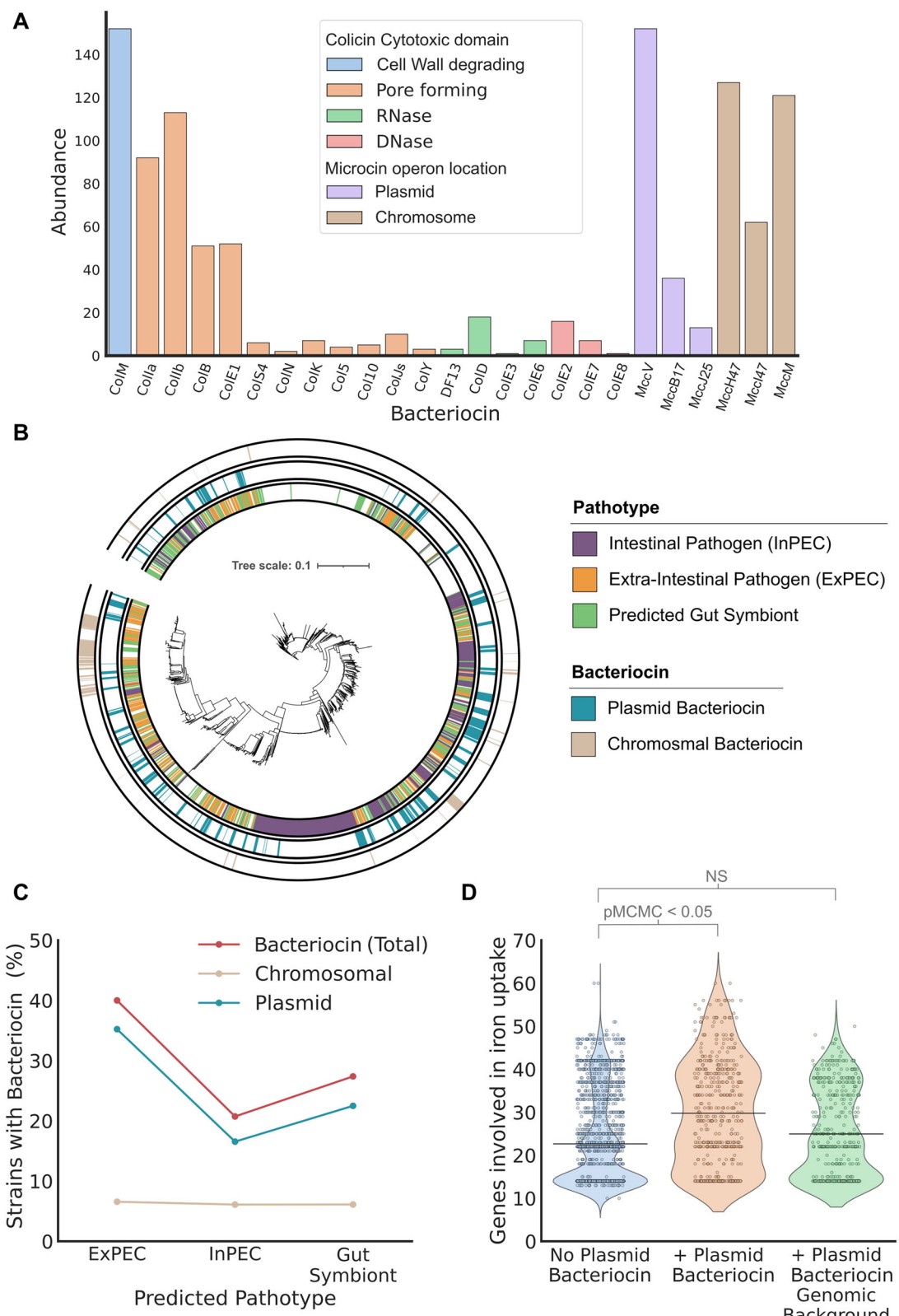

**Fig. 1 | The bacteriocins of *E. coli* are diverse and associated with particular pathotypes. A** Abundance of different bacteriocins found throughout 2601 *E. coli* genomes. **B** Core-genome phylogeny of 2601 *E. coli* strains with predicted pathotype (inner ring), presence of bacteriocin plasmids (middle ring) and chromosomal bacteriocins (outer ring). **C** Bacteriocin plasmids are not evenly distributed between non-pathogenic and pathogenic strains: bacterial genotypes that cause extraintestinal infections are particularly likely to carry a bacteriocin. **D** Bacteria with iron-uptake targeting bacteriocin plasmids are enriched in iron-uptake genes (orange) compared to strains without bacteriocin plasmids (blue) (MCMCglmm, posterior $\mu = 6.07$, 95% CI = [5.500, 6.669], pMCMC < $3 \times 10^{-4}$). Removing genes encoded on these plasmids (green) reveals that that the enrichment is driven by the fact that bacteriocin plasmids carry iron-uptake genes rather than the plasmids associating with genomes that carry high numbers of iron uptake genes (MCMCglmm, posterior $\mu = 0.4$, 95% CI = [−0.1709, 0.9314]), pMCMC = 0.15).

patterns are striking given that only around 20% of *E. coli* bacterial strains have a bacteriocin at all. More generally, we find that strains with multiple weapon systems occur in much greater numbers than predicted if weapons were distributed independently (Poisson distribution KS test $p < 0.005$), with 3.5% of strains (93 strains) encoding more than one weapon, compared to 0.83% (22 strains) predicted using a Poisson distribution (Supplementary Fig. 1). In sum, these analyses suggest that there is a subset of *E. coli* strains that are particularly specialised for bacterial warfare.

## Bacteriocin plasmids are enriched in extra intestinal *E. coli* infections

*E. coli* strains are often harmless symbionts that colonise the gut microbiome and can even protect against incoming bacterial pathogens[10,13,43,44]. By contrast, some *E. coli* strains cause severe intestinal and extra-intestinal disease—including: urinary tract infections, sepsis, pneumonia and meningitis – to the extent that *E. coli* as a species is a dominant cause of mortality[45], and specifically the mortality associated with antibiotic resistant infections[46]. As a result, there is great interest in the drivers of *E. coli* virulence.

Many intestinal *E. coli* pathotypes are clinically defined by the presence of a small number of virulence factors[47–49], while extra-intestinal infections are defined from the presence of *E. coli* in host niches outside the gut e.g. urinary tract, bloodstream or lung. We can, therefore, use the presence of certain virulence factors, alongside metadata on where a strain was isolated from, to categorise our strains of *E. coli* into Intestinal Pathogenic (InPEC), Extraintestinal Pathogenic (ExPEC), or gut associated and lacking virulence factors, which we use as a proxy for non-pathogenic *E. coli* gut symbionts (Table 2). This classification is reliant upon the quality of the available metadata and does not distinguish between hybrid strains capable of causing multiple pathologies. Nevertheless, across the large dataset we have analysed, we see clear and interesting differences between the strain categories. Our analysis also agrees with known trends in *E. coli* phylogroups and sequence types. We find that phylogroup B2 is the most common group in ExPECs, E and B1 in InPECs and A in predicted gut symbionts which agrees with PCR and genomic analysis of *E. coli* pathotypes[50–52] (Supplementary Fig. 2). Our methods also correctly classified common clinically important ExPEC lineages such as ST95, ST69 and ST131 and InPEC lineages such as ST11[53] and ST21[54].

Our analysis reveals significant differences in the carriage of bacteriocins by the different pathotypes (Fig. 1C), with bacteriocins significantly enriched in the ExPEC pathotype compared to other strains (binaryPGLMM $\beta = 0.60$, SE = 0.20, $p < 0.005$). Splitting bacteriocins into plasmid and chromosomal makes it clear that it is the plasmids that drive the association between bacteriocins and ExPEC strains (Plasmid weaponry: binaryPGLMM $\beta = 0.93$, SE = 0.22, $p < 0.001$, Chromosomal weaponry: binaryPGLMM $\beta = -0.14$, SE = 0.41, $p = 0.74$) (Fig. 1C). Consistent with this pattern, ExPEC strains are dominated by globally disseminated pandemic lineages and bacteriocin plasmids are prevalent in a number of clinically-important ExPEC sequence types including: ST95 (52% of strains), ST58 (50%), ST69 (27%), ST131 (25.6%) and ST457 (28.6%).

## Table 2 | Virulence factors used to distinguish between different *E. coli* pathotypes

| Virulence factor Marker | Description | Pathotype |
|---|---|---|
| *eae* | Intimin attachment to epithelial cells | EPEC[141] |
| pINV | Invasion plasmid | EIEC[142] |
| LT | Heat labile enterotoxin | ETEC[143] |
| ST | Heat stable enterotoxin | ETEC[144] |
| Stx | Shiga toxin | EHEC[145] |

We next sought to identify the mechanisms that explain the link between bacteriocin plasmids and disease. Looking for associations across a range of *E. coli* virulence factors suggests that one class of virulence factors is key to the association with bacteriocins: mechanisms of iron acquisition (MCMCglmm, posterior $\mu = 6.07$, 95% CI = [5.500, 6.669], pMCMC < 0.01, other virulence factor classes were not significant) (Fig. 1D). This pattern could be explained by the bacteriocin plasmids encoding iron-uptake virulence factors or the plasmids occurring in strains that encode iron-uptake genes in their chromosome. However, we found no correlation between plasmid-borne bacteriocins and chromosomal iron-uptake genes (MCMCglmm, posterior $\mu = 0.4$, 95% CI = [−0.1709, 0.9314]), pMCMC = 0.15). Instead, the data suggest that plasmids can encode both bacteriocins and virulence factors and bring them into a cell together. We, therefore, decided to focus on the gene content of the plasmids to explore this pattern further.

## Bacteriocin plasmids are enriched in siderophores and other virulence factors

To further investigate the traits carried by bacteriocin plasmids, we turned to a large plasmid database. Scanning 5373 *E. coli* plasmids from the PLSDB[55] database reveals 660 plasmids encoding bacteriocins (12% of all *E. coli* plasmids in PLSDB: 568 colicin plasmids and 199 microcin plasmids). Consistent with the above analyses, these bacteriocin-carrying plasmids are enriched in genes related to siderophores and the acquisition of iron, even after adjusting for plasmid size (GLM, $\beta = 0.06$, $p < 2 \times 10^{-16}$) (Fig. 2A, B) However, the associations with siderophores only occur for certain bacteriocins, ColM, ColIa, ColIb, ColB, MccV and MccJ25, which all target iron-uptake receptors (CirA, FepA, FhuA), (Fig. 2A, C). Given that their target receptors are involved in the uptake of siderophores and iron, hereafter we call these bacteriocins 'iron-targeting bacteriocins' as an abbreviation of iron-uptake receptor targeting bacteriocins. Not only are iron-targeting bacteriocins associated with siderophores, the plasmids that carry siderophores often encode multiple iron-targeting weapons, including combinations of both microcins and colicins. For example, microcin V is associated with an iron-targeting colicin on 66% of MccV plasmids that also all carry siderophores.

The iron-targeting bacteriocin plasmids carry a range of siderophore systems: salmochelin and aerobactin, or ABC transporters such as Sit and Eit, all of which are important virulence factors involved in the acquisition and import of iron[56–59] (Fig. 2C). For example, salmochelin is a derivative of the siderophore enterobactin that is able to function in spite of the host releasing lipochalin-2 to suppress enterobactin functioning[60], and aerobactin is a citrate-hydroxamate siderophore with extremely high affinity for iron[58]. Colicins are upregulated by DNA damage and we see this in our dataset, with LexA boxes upstream of nearly all of colicin genes (Supplementary Fig. 3). However, in addition to LexA, we find the majority of the iron-targeting bacteriocins associated with siderophores are upregulated via the Fur system, which responds to low-iron concentrations[61]. Fur regulated genes also include siderophores and their receptors, including the receptors targeted by iron-targeting bacteriocins: Cir[62], FepA[63] and FhuA[64].

In addition to siderophores, we find a range of other virulence factors encoded on the plasmids that contain iron-targeting bacteriocins (Fig. 2C, Table 3), including temperature sensitive hemagglutinin[65,66], hemolysin[67] and haemoglobin-binding protease[68] which have all been implicated in the virulence of extra-intestinal *E. coli*. We also identify iron-targeting bacteriocin plasmids which lack siderophores but encode virulence factors such as adhesins, fimbriae, enterotoxins and proteases associated with many intestinal pathotypes including enterotoxic, shiga-toxic and enterohemorrhagic *E. coli*. These non-siderophore plasmids (Fig. 2C) can encode three different

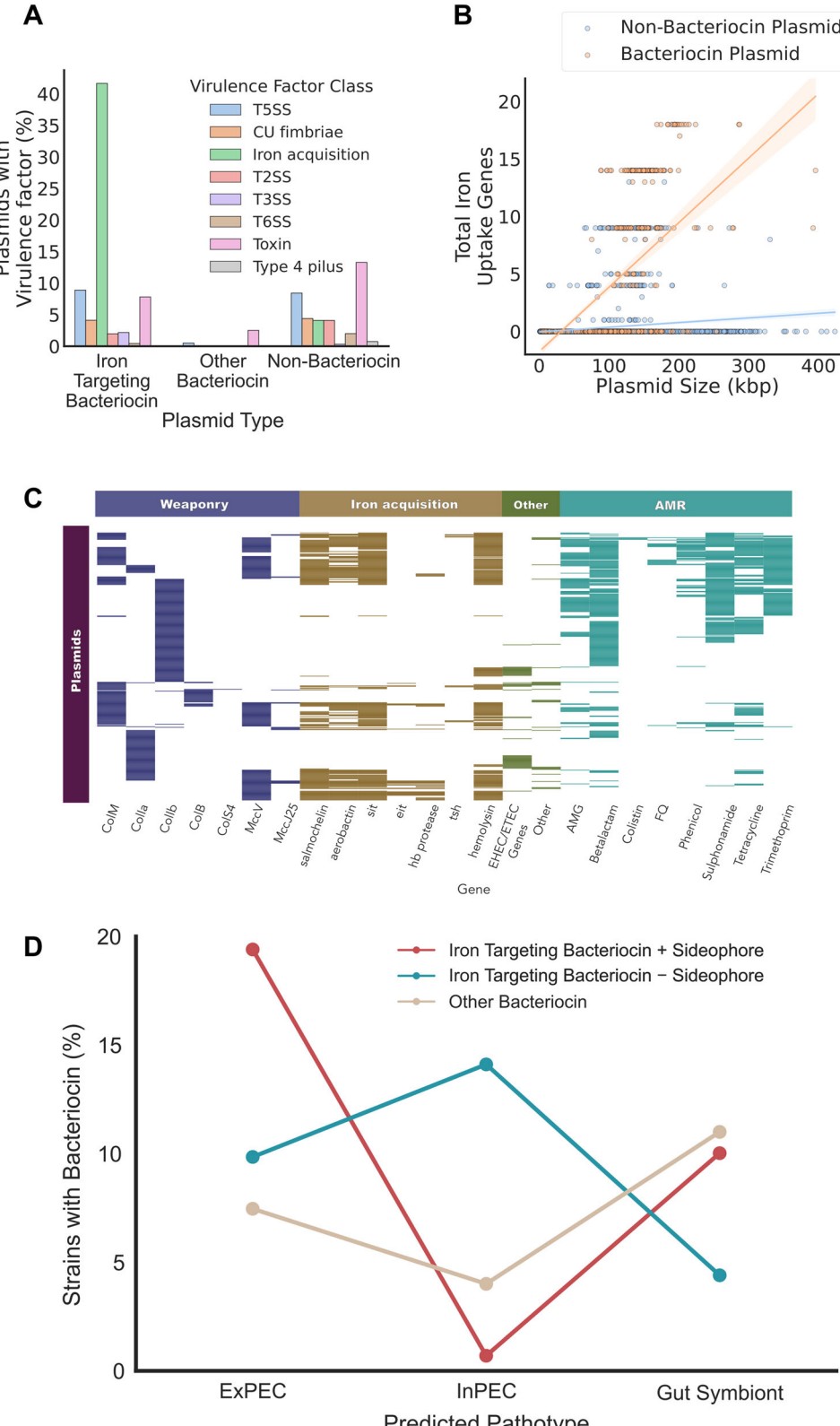

**Fig. 2 | Bacteriocin plasmids carry a range of virulence factors and AMR genes and are enriched in iron-uptake genes. A** Bacteriocin plasmids are enriched in genes associated with iron-uptake genes as compared to other plasmids. **B** Bacteriocin plasmids are more likely to encode iron-uptake genes than non-bacteriocin plasmids, even when accounting for plasmid length (GLM, $\beta = 0.06$, $p < 2 \times 10^{-16}$). Shaded areas indicate 95% CIs. **C** Iron-targeting bacteriocins occur alongside diverse iron acquisition systems and other loci involved in extraintestinal, enterohaemorrhagic and enterotoxic pathotypes. Bacteriocin plasmids can also carry antibiotic resistance genes for a wide range of antibiotic classes. **D** Different *E. coli* pathotypes tend to carry different types of bacteriocin plasmid. *E. coli* strains that cause intestinal infections often carry plasmids with iron-targeting bacteriocin but without siderophores, while ExPEC strains are enriched for all types of iron-targeting bacteriocin plasmids.

**Table 3 | Virulence factors associated with bacteriocinogenic plasmids from Intestinal pathogens**

| Virulence Factor | Description | Strain references |
|---|---|---|
| **Group 1** | | |
| *espP* | Serine Protease, which cleaves coagulation factor V[146] | Outbreaks of haemorrhagic colitis in Montana in 1994[147] and from Haemolytic Uraemic Syndrome (HUS) patients in Australia[148] |
| *epeA* | serine protease that degrades mucin[51,149] | |
| Saa | autoagglutinating adhesin[150] | |
| *subAB* | subtilase cytotoxin[151] | |
| **Group 2** | | |
| ST | heat-stable enterotoxin[152] | Shiga toxin encoding *E. coli* ST443 found in cattle from south America[153] and cases of diarrhoea[154] |
| Csf | CS5 fimbriae[155] | |
| Css | CS6 fimbriae[156] | |
| *aat* | anti-aggregation protein or dispersin[157] | |
| *pic* | mucin degrading enzyme[158] | |
| **Group 3** | | |
| *nleA* | immune disrupting effector of the T3SS[159] | *E. coli* O55:H7 strains, a progenitor strain to the highly pathogenic *E. coli* O155:H7, but itself responsible for major diarrhoea outbreaks and cases of HUS[160] |
| *stcE* | mucin protease[161] | |
| Etp | T2SS[162] | |
| *ibeC* | outer membrane protein linked to epithelial adherence and meningitis[163] | |
| *katP* | catalase-peroxidase[164] | |

sets of virulence factors (Table 3) and many have been linked to outbreaks of severe intestinal disease and HUS.

We also find that iron-targeting bacteriocin plasmids encode large arrays of AMR genes: 62% of all iron-targeting bacteriocin plasmids contain AMR genes predicted to provide resistance to at least one class of antibiotic, with 36.7% encoding resistance to ≥3 antibiotic classes. In contrast, only 9% of non-iron targeting bacteriocin plasmids encode AMR genes. Mechanisms of resistance to 8 different antibiotic classes are seen in the data, including resistance to the 'last resort'[69] antibiotic colistin, and one bacteriocin plasmid is predicted to have resistance to all 8 classes: pAR349, which encodes the ESBL $bla_{CTX-M-14}$ and colistin resistance gene *mcr-1* (Accession: NZ_CP041997.1). The combination of bacteriocins and AMR genes was not restricted to a single replicon but found on 42 different replicons (predicted by PlasmidFinder). The most common bacteriocin/antibiotic combination replicons predicted were IncI1-Iγ, which encodes genes for ColIa/Ib, followed by different IncF replicon subtypes. Bacteriocin carrying plasmids are not more likely than other plasmids to carry AMR genes once one accounts for their large size (GLM, $\beta = -0.0001$, $p = 0.34$) (Supplementary Fig. 4). Nevertheless, the effect is that, when a bacterial strain carries one of these plasmids, it becomes more likely to carry AMR genes than other strains, meaning that weaponry and AMR are significantly associated in *E. coli* (MCMCglmm, posterior $\mu = 0.9138$, 95% CI = [0.6896, 1.1461]), pMCMC = <0.01) (Supplementary Fig. 5).

### *E. coli* bacteriocins associate with different pathotypes based on receptor target

Our work suggests that large iron-targeting bacteriocin plasmids drive the key association between bacteriocin use, virulence and AMR in *E. coli*. These plasmids bring with them a complex of phenotypes. In particular, these plasmids often appear to help bacteria compete under low-iron conditions, such as during inflammation, when both iron-acquisition systems and iron-targeting weapons can be useful, as has been shown for ColIb[20] and certain microcins[14].

Consistent with the importance of these large iron-targeting bacteriocin plasmids, we do not observe any association between non-iron targeting bacteriocin plasmids and bacterial pathotypes in our dataset of *E. coli* genomes. By contrast, the iron-targeting bacteriocins are significantly more common in ExPEC vs non-pathogenic strains (Iron targeting bacteriocins: binaryPGLMM $\beta = 1.02$, SE = 0.23, $p < 0.001$). However, there are also differences among pathogenic

strains in the bacteriocin plasmids that they carry. ExPEC are much more likely to carry plasmids with both iron-targeting bacteriocins and siderophores (binaryPGLMM $\beta = 3.18$, SE = 0.72, $p < 0.001$) as compared to InPEC, which tend to carry iron-targeting bacteriocin plasmids lacking siderophores (binaryPGLMM $\beta = 1.06$, SE = 0.44, $p < 0.05$). The specific genetic repertoire of bacteriocin plasmids is, therefore, associated with different *E. coli* pathologies, with siderophores associated with ExPEC strains while host-targeting toxins are abundant in InPEC strains (Fig. 2D). Given the importance of intestinal carriage for all *E. coli* strains, it is likely that these patterns reflect differences in the ecology of ExPEC and InPEC strains in the mammalian gut. However, the extra siderophores carried by ExPEC strains may also be important for their tendency to disseminate to other body sites[70–72].

Our analyses fit with previous work on *E. coli* bacteriocins. For example, plasmids encoding the microcin MccV (known as pColV plasmids) have been previously identified in ExPEC strains and found to be maintained in lineages for long periods[73,74]. We also found pColV plasmids to be common in ExPEC strains, with 15.8% of predicted ExPEC strains encoding a MccV+ plasmid[74]. Previous research also has shown that ExPEC strains commonly possess plasmids with specific IncF replicons[73–75]. Analysing our ExPEC strains with pColV plasmids, we find the commonly reported IncF replicon sequence types including F18:A-:B-, F18:A-:B1, F24:A-:B1, F24:A-:B- and F2:A-:B1. In addition, we found pColV plasmids commonly encode multiple bacteriocins, with 58% of pColV plasmids possessing additional bacteriocins such as ColM, ColIa and ColIb. Moreover, other ExPEC strains in the dataset encoded ColIa, ColIb and ColM bacteriocins on a range of MccV- IncF and IncI plasmids.

### Colicin and microcin plasmids are diverse and mobilisable

Our work suggests that the weaponry, virulence factors and AMR genes encoded on bacteriocin plasmids are important for the ecology and virulence of *E. coli*, yet we know little about these plasmids in the context of the full set of *E. coli* plasmids (its plasmidome). To study these plasmids, we can use PlasmidFinder[76] annotations provided in PLSDB to categorise plasmids by their replicons (regions of DNA associated with plasmid replication). This analysis reveals 93 different replicons across 5373 *E. coli* plasmids and, whilst bacteriocin-carrying plasmids represented only ~12% of our dataset, they carry over a third of all plasmid replicons found in *E. coli* (32 replicons). Even individual

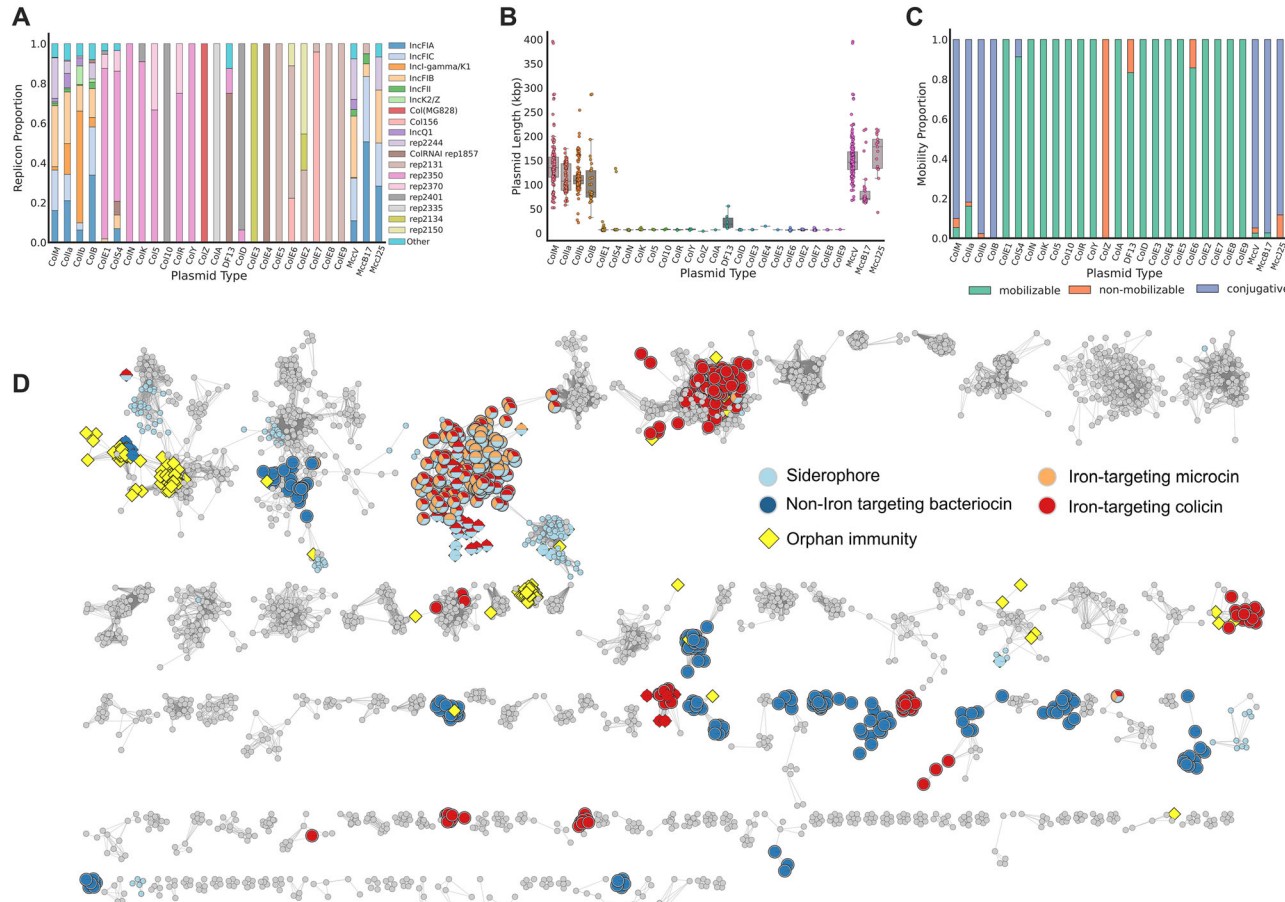

**Fig. 3 | Bacteriocin plasmids are diverse in size, replication machinery and mobilisation, and orphan immunity genes occur widely. A** Bacteriocin genes occur alongside a wide range of replicon types. The 19 most abundant replicons out of a total of 31 are shown. **B** Bacteriocin plasmids vary in size and broadly divide into small and large plasmids. Each point represents a single plasmid. Plasmids with multiple colicins are represented in multiple columns, total colicins = 880. Lower and upper hinges indicate the 1st and 3rd quartiles. Central lines indicate the median. Whiskers indicate the furthest point within 1.5× the interquartile range. **C** Bacteriocin plasmids are commonly mobilisable with larger plasmids encoding conjugation machinery and smaller plasmids carrying the genes to mobilise via a helper plasmid. **D** Clustering of 5353 *E. coli* plasmids from PLSDB. Bacteriocin plasmids are diverse and spread throughout the *E. coli* plasmidome. Each circle represents an individual plasmid, edges indicate a Mash distance below the cut off (0.03). Only plasmid communities which contain ≥ 5 members are represented.

colicins were encoded on plasmids with a range of different replicon types. For example, colicin M is found with 16 different replicon types, whilst colicins Ia and Ib both associate with more than 10 different replicon types (Fig. 3A). More generally, bacteriocin plasmids are commonly mosaic plasmids in the sense that one can find multiple replicon types on a single bacteriocin plasmid (282/660 bacteriocin plasmids have multiple replicons, mean number of replicons per bacteriocin plasmid = 1.7). These patterns suggest frequent genetic exchange among the bacteriocin plasmids, including the transfer of the replication machinery itself. Carrying multiple replicons in this way may allow for a broader host range for plasmids, because replication machinery can differ in its ability to function in different bacterial strains and species[77].

Previous analysis of colicin plasmids grouped them into type I small (~10 kb) multicopy plasmids and type II large (~40 kb) monocopy plasmids[31]. However, this distinction is based on small numbers of archetypal plasmids. We, therefore, looked in the large plasmid dataset to see if colicin plasmids lengths fell into these groups. This analysis reveals that, whilst there was a clear distinction between small type I and large type II bacteriocin plasmids, many far exceeded the 40 kb size estimate described in literature (Fig. 3B), with both colicin and microcin plasmids showing a large variation in size. For example, colicin M plasmids show a ~7.5-fold range in plasmid size (min = 52,297 bp, max = 395,758 bp).

Plasmid mobility - whether they can transfer to other cells via conjugation - has the potential to have a major impact during bacterial warfare because a weapon-encoding plasmid moving to a new strain could render it immune to attack[75,78–81]. Plasmid mobility can be predicted using the tool Mob-Suite[82]. Strikingly, we find that over 97% of colicin plasmids are predicted to be mobile, with 62.8% encoding their own conjugation machinery and a further 34.7% mobilisable by a helper plasmid (Fig. 3C). Similarly, 95% of microcins were mobilisable by conjugation. This analysis suggests that bacteriocin plasmids are highly mobile, even as compared to the average plasmid in the *E. coli* population where only 78% were predicted to be mobile.

## Bacteriocin plasmids and orphan immunity proteins occur throughout the plasmidome

The number of replicon types in bacteriocin-carrying plasmids suggests that they are phylogenetically diverse. Classifying diverse plasmids is difficult; recombination, insertions and co-integration means plasmid evolution cannot be explained with typical phylogenetic techniques. However, the use of approaches based on sequence similarity allows one to group plasmids into 'communities', which are clusters of related plasmids which share large percentages of DNA[83]. Using this approach, we are able to assign 73.7% of all *E. coli* plasmids into clusters with 5 or more members (Fig. 3D and Supplementary Fig. 6). Clusters are well defined and mixing between clusters is rare

(assortativity coefficient (a measure of edges shared between clusters, with 1 indicating no shared edges): 0.998). Consistent with our earlier replicon analyses, we find bacteriocin plasmids in 96/1148 plasmid clusters and they are well-dispersed across the *E. coli* plasmidome. Type I colicin plasmids tend to form clusters with plasmids encoding the same or closely related colicins. The larger type II plasmids exist in more diffuse clusters, which contain colicin, microcin and non-weaponry plasmids. We observed no mixing between iron-targeting bacteriocins and non-iron-targeting bacteriocins (assortativity coefficient 1), suggesting a limited shared evolutionary history. Interestingly, there are also many clusters where a single colicin plasmid occurs with non-colicinogenic plasmids implying a recent horizontal acquisition of a colicin gene from another plasmid.

Consistent with its functional importance, the association between bacteriocins and siderophores is not confined to a single plasmid cluster in our network analysis. Association of salmochelin and bacteriocins is observed in 14/19 clusters where salmochelin was present. More generally, salmochelin, aerobactin, Eit and Sit iron-uptake systems, are all significantly associated with clusters that encode iron-targeting bacteriocin plasmids (Fisher test: salmochelin: OR = 69.05 $p = 1.34 \times 10^{-15}$, aerobactin: OR = 7.28 $p = 6.62 \times 10^{-4}$, eit: OR = 16.71 $p = 1.1 \times 10^{-4}$, sit: OR = 15.71, $p = 1.46 \times 10^{-8}$).

Colicins, and most bacteriocins, are typically encoded as a toxin and downstream immunity protein. However, orphan immunity genes without a cognate toxin can protect a cell from incoming attacks without suffering the high cost of toxin production[84]. We find 243 plasmids that encode an orphan immunity gene, with more colicin immunity proteins than those for microcins (195 vs 53), which likely reflects the higher prevalence of colicins. Indeed, as is expected, the types of immunity orphan gene observed mirror the most common toxin domains carried by strains (Pore forming: 146, Microcin: 53, Peptidoglycan inhibiting: 46, tRNase: 2, DNase: 1). It is not known if orphan immunity genes occur due to natural selection against the high cost of toxin production (toxin loss) or because immunity genes are naturally selected to provide protection against toxin-wielding competitors (immunity gain). Distinguishing between toxin loss and immunity gain is important, because it can help decipher the role of bacteriocins in bacterial ecology. If bacteriocins evolve only because they can function as toxin-antitoxin systems that help to maintain plasmids, one expects to only find orphan immunity genes alongside recently degraded toxin genes within plasmid clusters containing bacteriocin plasmids (toxin loss). If bacteriocins are important agents in bacterial warfare, we would also expect cases where immunity genes are present in clusters without toxin genes, because picking up and maintaining immunity genes will confer protection from attacking bacteria (immunity gain).

136 of the orphan immunity plasmids that we observe occur in plasmid clusters which do not contain a bacteriocinogenic plasmid (across 24 distinct clusters). This pattern supports the immunity gain hypothesis and suggests that bacteriocin immunity, and therefore bacterial warfare via bacteriocins, is important for *E. coli*. One potential limitation of this analysis would be if we are missing the bacteriocinogenic plasmids that occur alongside orphan immunity plasmids due to insufficient sampling. However, there are many examples of clusters that contain multiple orphan immunity plasmids (range 2–58 orphan immunity plasmids per cluster) but no bacteriocinogenic plasmids, which is difficult to explain by sampling limitation (Fig. 3). Moreover, inspecting coding sequences from the four largest clusters of orphan immunity plasmids provides further evidence that the occurrence of orphans is not simply explained by them having a recently degraded toxin gene (Supplementary Fig. 7). The plasmids examined either completely lack a bacteriocin gene or they possess a heavily truncated bacteriocin gene, where this same truncation is found in multiple distinct plasmids in a cluster i.e. it has been maintained in plasmids of different sizes that were isolated from different locations (Supplementary Fig. 8). These patterns, therefore, all support the idea that the orphan immunity genes are being independently maintained to protect against the toxins of other strains.

To further cement the conclusion that these bacteriocins are not simply plasmid addiction systems, we also analysed the plasmids for the presence of other toxin-antitoxin (TA) systems, which act inside the cell, and are known to be able to function as plasmid addiction systems, and are certainly not involved in bacterial warfare. We reasoned that, if the only function of bacteriocins is to stabilise plasmids, one expects fewer canonical toxin-antitoxin addiction systems on bacteriocin plasmids than other plasmids. Both presence and number of TA systems scales with plasmid size (GLM: Presence of TA system (binary) ~ Plasmid size (kbp), $\beta = 0.0032$, $p < 0.01$; TA system count (continuous) ~ Plasmid size (kbp), $\beta = 0.0084$, $p < 0.01$). However, the presence of a bacteriocin does not lower the probability of a TA system being on the same plasmid when controlling for plasmid size. In fact, non-bacteriocin plasmids have fewer canonical toxin-antitoxin systems than the bacteriocin plasmids, when accounting for plasmid size (Supplementary Fig. 9).

## Colicin and microcin genes are mobile among plasmids

Many bacteriocins are found in multiple clusters across our plasmid network, which suggests that either bacteriocin plasmids have evolved and diversified or bacteriocin genes are moving between plasmids. To better understand the processes at play, we built a phylogeny of one of the common colicins (ColIa), which identifies multiple transitions between plasmid clusters (Fig. 4A). Comparing the plasmids associated with these transitions reveals examples where colicins sit in a large region that is shared between plasmids in different clusters, which is consistent with bacteriocin plasmids evolving and diversification. However, we also see cases where there is no significant region (>2kbp) of homology between plasmids (other than a shared colicin gene), which suggests that bacteriocins alone are moving between plasmids. Further consistent with this inter-plasmid mobility, many bacteriocin genes are flanked by two transposons which have the potential to transfer at the same time (i.e. composite transposons) and transport the DNA segment between them, including ColM, ColIb, ColIb, CloDF13, MccV, MJ25 and MccB17 (Fig. 4B). Capturing transposon movement between plasmids is difficult due to insufficient sampling depth and lack of longitudinal sampling. Despite this, we were able to identify events where a predicted composite transposon was shared to a high identity between two unrelated plasmids, which again suggests that the bacteriocins are moving between plasmids via composite transposons (Fig. 4C).

## Colicin plasmids are shared amongst Enterobacteriaceae

Unlike microcins, colicin-like proteins require no complex post-translational modifications or dedicated export system. This simplicity means that colicins have the potential to function in a wider range of species than microcins, and colicins are found throughout Enterobacteriaceae[26,85]. However, the degree to which colicin plasmids are shared between species and genera is not well understood. To investigate this, we used the same network-based approach as before to cluster the 11274 plasmids from 5 genera of *Enterobacteriaceae* (*Escherichia, Klebsiella, Salmonella, Citrobacter* and *Shigella*), in the plasmid (PLSDB) database (Fig. 5A). 8373 of these plasmids (74.3%) can be placed within one of 260 plasmid clusters that have five or more members (the assortativity coefficient of 0.998 again suggests well defined clusters). Many of these clusters contain plasmids from multiple genera (131/260), and edges between genera are common in the network (assortativity coefficient = 0.43), which is consistent with general ability of plasmids to move within *Enterobacteriacae*. We then identified all colicins and related bacteriocins across *E. coli, Citrobacter, Shigella* (colicins), *K. pneumoniae* (klebicins) and *S. enterica*

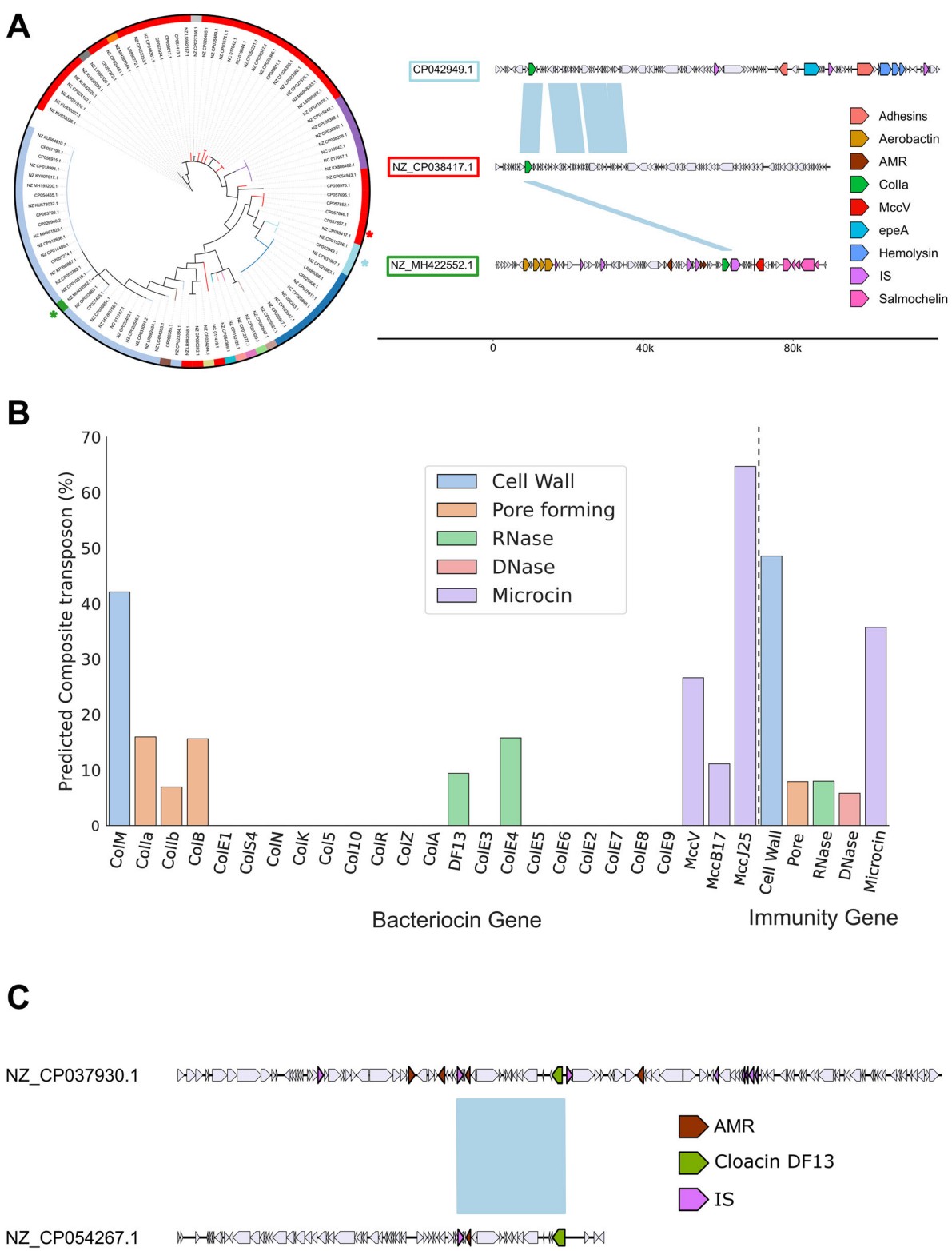

**Fig. 4 | Specific bacteriocin genes are mobilisable and found across diverse plasmids. A** Phylogeny of colicin Ia genes (*cia*-toxin and *iia*-immunity gene) with the plasmid communities they were identified with (left). Alignment of three ColIa plasmids shows a high level of diversity between plasmids, where either a small region of DNA or even just the ColIa genes (*cia* and *iia*) are shared between

plasmids (Regions of similarity > 2000bp are shown in blue). **B** Percentage of bacteriocin (or immunity) genes predicted to occur within a composite transposon. **C** Example of an alignment of two colicin plasmids that share a shared predicted composite transposon that carries a colicin.

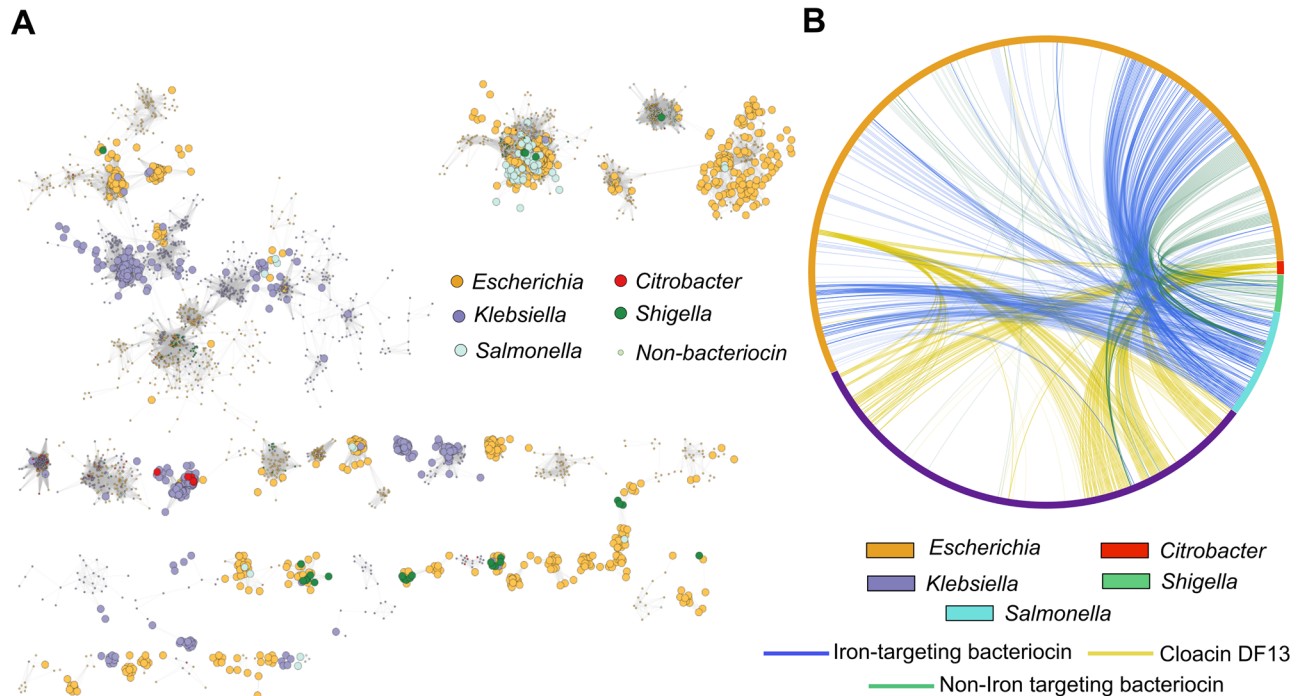

**Fig. 5 | Colicin plasmids are shared among multiple genera within *Enterobacteriaceae*. A** Louvian clustering of all plasmids in the genera *Escherichia*, *Klebsiella*, *Salmonella*, *Citrobacter* and *Shigella*. For clarity, only plasmid clusters with contain at least 1 colicin plasmid are shown. Each circle represents an individual plasmid and edges indicate a mash distance of < 0.03. Non-colicinogenic plasmids are shown as smaller circles. **B** All colicin plasmids identified in 5 different genera of Enterobacteriaceae are represented in a chord diagram. Connections show plasmids with ≥ 99% ANI shared between genera.

(salmocins) in the plasmid (PLSDB) database and mapped them to the clusters.

This analysis shows that 52 clusters contain at least one colicin plasmid and over 50% (28) of the colicin clusters contained plasmids from more than one genus (assortativity coefficient = 0.36). However, different colicin plasmids are distributed differently between the genera. The edges between *Escherichia, Klebsiella* and *Citrobacter* colicin plasmids are mostly between plasmids encoding Cloacin DF13. This bacteriocin hijacks the siderophore receptor IutA[86] of the aerobactin system, a siderophore system that is found in many *Enterobacteriaceae*, including *E. coli* and *Klebsiella* where it is an important virulence factor[87]. Edges between *Klebsiella* and other genera (*Salmonella* and *Shigella*) or involving non-Cloacin DF13 plasmids are relatively rare. Supplementary analyses suggest that the low rate of plasmid transfer with *Klebsiella* is not due to an inability to share plasmids between the species as closely related non-bacteriocin plasmids are found in *Salmonella* and *Klebsiella* strains (Supplementary Fig. 10). By contrast, edges between *Escherchia* and *Salmonella* colicinogenic plasmids are common, typically involving ColIa and ColIb plasmids and comprised 80% of all edges between species in our analysis (Fig. 5B and Supplementary Fig. 11). A dedicated phylogeny of ColIa toxin and immunity genes from *E. coli* and *S. enterica* also suggests a high rate of transfer between these species (Supplementary Fig. 12). These patterns may reflect the niche in which *Salmonella* will often encounter *E. coli*, the inflamed mammalian gut where possession of an iron targeting ColIb plasmid can provide a fitness advantage to blooming enterobacteria[20,78].

## Discussion

Bacteria have evolved a diverse array of weapons that target competing strains, many of which have been extensively studied under laboratory conditions[7,17,42]. Here we have used the unrivalled genomic resources of *E. coli* to investigate the natural ecology and evolution of bacteriocins. The bacteriocins of *E. coli* occur both on the chromosome and on plasmids, and we find evidence that plasmid-encoded bacteriocins can readily move between both strains and plasmids. Many of the plasmids that carry bacteriocins also carry AMR genes and virulence factors. This result suggests that problematic strains are commonly using weapons in order to establish and persist in communities.

In particular, we find that bacteriocins are associated with siderophores and other factors associated with growth under low-iron conditions. This pattern is driven by five bacteriocins: ColIa, ColIb, ColM, ColB and MccV, all weapons which themselves target iron uptake receptors (CirA, FepA and FhuA). Iron-limitation is common in hosts, especially during inflammation[88]. Under these conditions, bacteria upregulate the expression of iron uptake receptors, which increases their susceptibility to iron-uptake targeting weaponry[20]. The use of such weaponry, therefore, may give strains a fitness advantage under low iron conditions[20], such as those found in the inflamed gut. From the gut, strains can then spread to extra-intestinal infection sights such as the urinary tract[89–91]. More generally, our findings suggest that iron-targeting bacteriocins, siderophores and other virulence factors can work in concert to help pathogens to both colonise diverse communities, such as the human microbiome and cause disease. Experimental tests are needed to confirm this hypothesis along with the potential for siderophores and iron-targeting bacteriocins to work in synergy to enable persistence of ExPEC strains in the microbiome[92–94]. However, consistent with these ideas, gut colonisation is an important step for the pathogenesis of extraintestinal *E. coli* in animal studies and extra-intestinal virulence factors can increase survival in the gut[70,71,92–97]. Disruption of gut colonisation represents a potential therapeutic strategy to prevent extra-intestinal *E. coli* infections[98]. Our work suggests that understanding the links between bacterial warfare and virulence will aid in this endeavour.

The function of bacterial weapons under natural conditions is surprisingly poorly understood. Though they are often potent killers under laboratory conditions, the in vivo activity of bacteriocins is less clear[16,99–101]. This pattern has led some to suggest that colicins function solely as selfish genetic elements, which prevent their loss from

bacterial hosts by intoxicating cells that lose them[24,25]. While colicins may indeed reduce plasmid loss from bacterial lineages[24], our analyses do not support the view that this is their primary function. We found evidence that orphan immunity proteins are being maintained throughout the *E. coli* plasmidome, which is consistent with natural selection for protection against bacteriocin attack. In addition, we find that conventional toxin-antitoxin systems are abundant on bacteriocin plasmids, which again suggests that the bacteriocins are performing a separate function.

Our analyses, therefore, support the intuitive idea that bacteriocins function as weapons for bacteria to compete with other strains, albeit in a manner that can also benefit plasmids that carry them[81]. We also find that that carriage of these weapons in *E. coli* is associated with particular ecologies, with a strong link between the common iron-receptor targeting colicins and virulence factors that help bacteria to cause disease. This link to virulence is further cemented by the observation that the bacteriocin plasmids of *E. coli* are commonly shared with the pathogen *S. enterica*. Overall, our work suggests that the bacteriocins of *E. coli* are both highly mobile and important for the ecology of dangerous strains.

## Methods

### E. coli genome dataset

*E. coli* genomes were accessed from the BV-BRC[102] database (formally known as PATRIC). Genomes were selected to ensure high quality complete genomes (Genome Status: Complete, Genome Quality: Good). Genomes of poor quality were removed by three methods. Firstly, any genome with an L50 > 1 (indicating the chromosome is split over multiple contigs). Secondly, the number of coding sequences versus the length of the genome was plotted. For an individual species this should be a linear relationship. We removed any genomes that deviated from the mean by >500 coding sequences[103] (Supplementary Fig. 13a). Finally, we used the panaroo-qc tool in the panaroo software to remove genomes with suspected contamination[104]. This tool uses mash to identify distances between the dataset and a database of refseq genomes. Multidimensional scaling allows visualisation of genomes which form outliers and are therefore possible contaminates (Supplementary Fig. 13b). The remaining 2601 genomes included samples isolated from 74 different countries, across six continents, with no single country representing more than 20% of the data. Genomes were sampled over 118 years between 1884-2022 and included genomes from 846 Bioproject Accessions and 104 publications. The dataset includes 438 different sequence types with 196 sequence types represented by more than 1 sequence (Supplementary Fig. 14). Genomes covered all known *E. coli* phylogroups (A: 990, B1: 482, B2: 374, C: 93, D: 234, E: 301, E/I: 1, F: 84, G:28, I: 4, Unknown: 10).

The pangenome of the 2601 *E. coli* genomes was calculated using Panaroo[104]. As colicins are present in populations at a low frequency we used the sensitive clean-mode parameter and set the core genome as genes present in ≥99% of genomes. Core genes were aligned with MAFFT and phylogeny calculated using RAxML-NG[105] with the Generalised time reversible model and gamma model of rate heterogeneity.

Virulence factors were identified using Abricate[106] and a database of curated *E. coli* virulence factors from VFDB[107] and literature[108]. AMR elements were identified using Resfinder[109]. Serotype was predicted using abricate and the EcOH dataset[110]. Sequence type was predicted using the *mlst* software[111,112]. Phylogroup was predicted using the ClermonTyping tool[113]. All abricate searches required a coverage and identity >70%. Colicin and colicin immunity proteins were identified using BLASTp (Supplementary Tables 1 and 2) with open reading frames predicted using getORF[114]. An open reading frame was considered a colicin only with an *e* value < $5 \times 10^{-5}$ and coverage >80% of query length. Some colicins are very rare and though we they were included in our initial search, they were not identified even in our dataset of 2601 genomes (e.g. ColA, ColE4 and ColR, ColZ, ColE9).

Uropathogenic specific protein (Usp) shares close homology to colicins and was included in the search and hits to Usp were removed from analysis. Microcins were identified using a custom database of microcin genes and Abricate[106]. IncF RSTs were predicted using pMLST[115].

### Pathotype prediction of E. coli genomes

*E. coli* exists as a non-pathogenic member of the microbiome but can cause intestinal (InPEC) and extraintestinal (ExPEC) infections. Different pathotypes within InPEC infections can be defined by a selection of virulence factors, whereas ExPEC infections are defined by isolation point (e.g. urinary tract, bloodstream). We used the presence of specific virulence factors often used as PCR markers[48] to define InPEC infections (Table 2) and scanned available metadata to identify *E. coli* isolated from non-intestinal body sites.

### Phylogenetic comparative analysis

To investigate the relationship between iron-targeting bacteriocins and different virulence factors, we used a Generalised Linear Mixed Model (MCMCglmm[116]) using the core genome phylogenetic covariance matrix to control for sampling. Chains were run with 100,000 iterations with a 10,000 burn-in and 30-step thinning. Uninformative flat priors were used ($V = 1$, nu = 0.02). Associations between binary traits such as *E. coli* pathotype and plasmids, were tested using binary Generalised Linear Mixed Models with different pathotypes as a binary response variable using the binaryPGLMM function in the Ape[117] package (e.g. Formula = Pathotype ~ Iron targeting Bacteriocin Plasmid, phy = Core gene phylogeny).

### Plasmid database

Plasmid sequences were accessed from PLSDB v2020_11_19, which contains plasmid sequences from NCBI and INSDC based on previously defined criteria[118] The 5373 E. coli plasmids were annotated as *Escherichia coli* as the host species. Host species of each plasmid in the database was determined from NCBI metadata. Plasmid sequences < 1000 bp were removed from analysis. Plasmid mobility was predicted using the mobtyper tool within Mob-Suite[82].

### Identifying orphan immunity proteins

Bacteriocin immunity proteins are small proteins with homology to many antitoxins and enzymes. They are normally confirmed to be colicin immunity proteins by the presence of the cognate toxin, therefore making identifying orphan immunities difficult. To identify orphan immunity proteins, we first constructed a dataset of immunity protein genes ≤ 200 bp from a cognate toxin. We used this set of genes to scan against the PLSDB using blastp with strict parameters and only selected hits with a percentage identity >99%.

### Toxin-antitoxin system analysis

Hidden Markov Models for Toxin-Antitoxin systems were accessed from TASmania[119] and PLSDB protein open reading frames were scanned using HMMer3[120] with a *e* value cutoff of $<5 \times 10^{-5}$. A Toxin/Antitoxin system was defined as a toxin and antitoxin in contiguous open reading frames which had both passed the threshold. Associations between bacteriocins and Toxin/Antitoxin systems were performed using Generalised Linear Models and R.

### Plasmid network analysis

To form a plasmid network, 'distances' between plasmids were determined using Mash[121] with a sketch length of 5000, k-mer size of 13 and all other parameters set to default. To form a network, a threshold of 0.03 was applied to the mash distances with distances below this threshold forming the edges between plasmid nodes. Plasmid communities were estimated using the Louvian algorithm[122] to optimise network modularity[83]. The resulting network and communities were visualised using CytoScape[123].

## Gene ontology enrichment analysis

Gene ontology enrichment analysis was performed on the *E. coli* genome datasets. First, the pangenome created using panaroo was annotated using interproscan (v5.57-90.0)[124,125] and the CDD, Pfam, Superfamily and TIGRFAM databases to identify Gene Ontology terms. GO abundances were normalised by plasmid size (kbp). GO term distribution was modelled as a poison distribution. Significant differences in poisson means were calculated between bacteriocin and chromosomal and non-bacteriocin plasmids using R. *P* values were corrected using a Bonferroni correction.

## Plasmid backbone phylogeny

To generate a phylogeny of plasmid backbone genes for an individual plasmid cluster, we first calculated the pangenome of all plasmids within the cluster using panaroo, with the sensitive clean mode and the core gene cut off at 90% of plasmids. Core genes were then aligned using MAFFT[126] and phylogeny calculated using IQTree2[127] and ClonalFrameML[128] to account for recombination. The resulting phylogeny was visualised using iTOL[129]. Colicin phylogenies were calculated from start of the toxin to end of the immunity gene. Sequences were aligned using clustalo[130] and a phylogeny calculated using IQTree2.

## MEfinder

MobileElementFinder[131] was used to identify composite transposons on plasmids. Before scanning with MobileElementFinder, all plasmids were doubled (an extra copy of the sequence appended to the end) so composite transposons would be found across the plasmid break point. All PLSDB plasmids were then scanned for the presence of insertion sequences and transposons. Predicted transposons were then analysed for the presence of bacteriocins. Plasmids were visualised using gggenomes[132].

## Prediction of Fur and LexA binding sites

Alignments of *E. coli* Fur and LexA binding sites were accessed from CollectTF[133]. Binding sites within plasmids were identified using FIMO[134] of the MEME[135] suite. Fur and LexA regulation were determined by the presence of a predicted binding site ($p < 5 \times 10^{-5}$) within 200 bp of the bacteriocin gene start codon.

## Data availability

The genomic data used in this study is publicly available from BV-BRC using accessions listed in Supplementary Data 1. Plasmid data is available from PLSDB using accessions listed in Supplementary Data 2 and 3. Publicly available sequences used to identify bacteriocins from E. coli are listed in Supplementary Table 1. Publicly available sequences used to identify Colicin-like proteins from Escherichia, Klebsiella, Salmonella, Citrobacter and Shigella, are listed in Supplementary Table 2. Antimicrobial resistance genes and virulence factors were annotated using Abricate and the Resfinder (http://genepi.food.dtu.dk/resfinder) and ecoli_vf datasets, (https://github.com/phac-nml/ecoli_vf/). Hidden Markov Models for Toxin-Antitoxin Systems were accessed from TASmania (https://shiny.bioinformatics.unibe.ch/apps/tasmania/).

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

## Acknowledgements

Thank you to Elisa Granato, Olivier Cunrath, Erik Bakkeren and Sean Booth for feedback and comments. K.R.F. is funded by European Research Council Grant 787932 and Wellcome Trust Investigator award 209397/Z/17/Z.

## Author contributions

Conceptualisation, C.S. and K.R.F.; Methodology, C.S.; Investigation, C.S.; Writing—Original Draft, C.S. and K.R.F; Writing—Review and Editing, C.S. and K.R.F.; Funding Acquisition, K.R.F; Resources, C.S. and K.R.F; Supervision, K.R.F.

## Competing interests

The authors declare no competing interests.
