## [Transparent Peer Review file · Nature Communications]

Bacterial warfare is associated with virulence and antimicrobial resistance

Corresponding Author: Professor Kevin Foster

Version 0:

Reviewer comments:

Reviewer #1

(Remarks to the Author)

This manuscript explores the role of bacteriocins in *E. coli*, especially their association with virulence and antimicrobial resistance (AMR) in an evolutionary perspective. Authors performed a comprehensive analysis of public genomic databases, combining phylogenetics, plasmidomics, and statistical modeling to propose that bacteriocin-encoding plasmids are key players in ExPEC pathogenesis and horizontal gene transfer. Most importantly, this work found that bacteriocin could be transferred along with resistance genes and virulence genes among pathogens. The narrative and data analysis are compelling and wonderful, but the analyses are correlative without experimental support. It's apparent this work focuses on data analysis, but such design should be discussed and the limitation should be stated clearly. Other specific minor comments were suggested as follows:

Specific comments

Line 78: What's the diversity of bacteriocin genes? Are they functioning in the same role or have different functions based on mutations? This background information should be clarified before genome analysis.

Line 92-103: Authors should present the distribution of different bacteriocins in a better way. It's difficult to catch the main idea in this part. Do multiple weapons mean different bacteriocins or various mutants of the same weapon?

Line 104-105: The study's findings do not support this conclusion.

Lines 117–119, 482–490 The classification of strains into ExPEC, InPEC, and non-pathogenic is based on isolation site and the presence of a few virulence markers. However, hybrid strains and context-dependent virulence are common in *E. coli*. Justify the use of these categories and acknowledge their limitations. Consider complementing them with phylogroup or MLST-based clustering for more robust classification.

Line 143: Rephrase this section title.

Line 186: CTX-M-14 should not be in italic. Check gene names in the right format.

Line 218: In this section, types of bacteriocin plasmids should be demonstrated in a logical way. What's the most dominant AMR plasmid harboring bacteriocin? Such specific information is important than overall demonstration.

Lines 218–252: Claims about bacteriocin gene mobility and plasmid spread rely heavily on in silico tools (MOB-suite, PlasmidFinder), without experimental or historical data.

Line 258: Authors can use different techniques to clarify plasmids, and the common one is to utilize replicon genes to screen. In *E. coli*, there are many well-known plasmids, such as typical AMR plasmids IncC/IncX3/IncQ/IncI2, etc. Did authors analyze plasmids based on that standard method? Also, the genetic environments of bacteriocin genes could be shown a little bit.

Line 346: In this part, I do not see the common plasmids in Enterobacteriaceae. What's the reason?

Line 374: Collb and Collb plasmids? Furthermore, the conclusion stated in this sentence is not supported by the current data presented in Figure 5b.

Line 386: It seems that authors did not analyze chromosomal bacteriocin compared to plasmids.

Line 416: This research is a good analytical work, but experiments were not performed to support such results. This limitation could be presented as an important section to imply a few future directions.

Figure 1B The scale bar is missing from the phylogenetic tree.

Several figures or figure legends are overly simplistic, lacking necessary descriptions and statistical analyses of differences. Please review and revise accordingly.

Some references have formatting inconsistencies and are missing page numbers. Please check and revise accordingly.

From Ruichao Li

Reviewer #2

(Remarks to the Author)

In the manuscript "Bacterial warfare is associated with virulence and antimicrobial resistance," the authors analyzed prevalence of bacteriocins across *E. coli* genomes available in databases using bioinformatic tools. *E. coli* is a very heterogeneous species, and its evolution and ecology are not fully understood. Therefore, complementary analyses at the population level are crucial. However, I have concerns about the relevance and presentation of their results. Major comments:

1. The authors analyzed set of *E. coli* genomes with well-defined inclusion/exclusion criteria. However, there are missing epidemiological information about used strains. Particularly, prevalence of individual phylogroups, serotypes, and pathotypes has to be added. For example, how many ExPEC strains belonging to phylogroup B2 were included in the analyzed set. This characterization is important for interpretation of the findings (see below).
2. Similarly, the authors did not include a list of the analyzed bacteriocins in the methods section. The different bacteriocin subsets are presented across the figures. And the list of bacteriocins appears to be incomplete (e.g., bacteriocins A, E4, E5, E9, U, R, and microcin PDI are missing from Figure 1). Furthermore, there are errors in classifications – e.g., MccB should be MccB17, and Col J should be ColJs. The list of known bacteriocins is available in reviews (e.g., doi: 10.1080/14787210.2020.1816824, doi: 10.3389/fmicb.2020.586433).
3. Furthermore, the authors analyzed over 5,000 plasmids from a separate database. How were these plasmids assigned to specific *E. coli* genomes, and were unassigned plasmids omitted from the study? How many *E. coli* genomes did not contain plasmid(s)?
4. The reviewer's previous comments relate to the finding that bacteriocin production was observed in only 20% of the *E. coli* strains analyzed. This contrasts with numerous prior studies across diverse *E. coli* sets, which reported significantly higher levels of bacteriocinogeny (approximately 50%). Notably, the prevalence of bacteriocin genes reached up to 75% in ExPEC isolates. There are two potential explanations for this discrepancy. First, there may be a sampling bias, with the majority of analyzed strains originating from environmental conditions and laboratory and representing non-pathogenic isolates, which typically belong to phylogroups A/B1 and have previously been associated with lower bacteriocin levels. Another possibility is that some analyzed genomes was compromised by plasmid loss during DNA isolation, a common issue in older sequencing data, which likely comprises part of this study.
5. Another major issue is the limited novelty of this study, as numerous previous studies have experimentally demonstrated that *E. coli* bacteriocins are: i) encoded on plasmids and horizontally transferred among *E. coli* strains; ii) encoded on plasmids that frequently harbor diverse virulence factors, including adhesins, invasins, and iron-acquisition systems; iii) regulated via SOS-damage and iron limitation; and iv) associated with pathogenic *E. coli*, particularly strains belonging to the B2 phylogroup and responsible for extraintestinal infections in humans.
6. The authors performed a considerable set of bioinformatic analyses; however, most of the presented data lacks informativeness. For example, the authors present a scheme of plasmid clusters, but the exact names for individual clusters are missing.

Reviewer #3

(Remarks to the Author)

Reviewer #4

(Remarks to the Author)

This is an interesting and well analysed body of work that speaks to the functions of bacteriocins and their role in *E. coli* ecology. In the first instance it is great to see novel work being undertaken to study *E. coli* ecology as it is an area that has been somewhat neglected. The overarching conclusions is that bacteriocins are important for bacterial ecology, competition and virulence. The manuscript is well written and there are no typographic errors.

The major findings of this work are that: i) bacteriocins are mobile being facilitated in their movement by the actions of flanking IS elements (compound transposons) and on plasmids; ii) *E. coli* bacteriocins are abundant in pathogenic strains and linked to known virulence factors such as siderophores, adhesins and enterotoxins; and iii) plasmids containing orphan immunity proteins, which protect against bacteriocin attack without producing toxins, supporting the importance of bacteriocins for bacterial competition.

The authors conclude their abstract with the following: Our work suggests that the bacteriocins of *E. coli* are important antibacterial weapons for dangerous antimicrobial-resistant strains.

Major concerns:

The major weakness of this manuscript is that the authors could do a much better job relating their findings to current thinking regarding the ecology of *E. coli* particularly ExPEC as well as the role of bacteriocins (if any) among well described ExPEC plasmids in ExPEC disease. Moreover the authors should consider a case study in F plasmids (known for carriage of VAGs), adding F plasmid MLST replicon types, and compare their finding and how they impact existing and recent F plasmid clustering studies.

In this regard the authors could add a Table of the major *E. coli* Sequence types. This refers to the following statement (line79): Bacteriocins are also very widely distributed across *E. coli* strains, being present in all *E. coli* phylogroups (G (75%,

21/28), B2 (46.3%, 173/374) and B1 (34.4%, 166/482)), and >90% (38/44) of all sequence types (STs) which had more than 10 representatives (Figure 1a). The readership would like to know more about these (major) sequence types and Fig 1 or the manuscript more broadly does not provide any information about this.

Furthermore, the authors should consider relating their finding to the major ExPEC F plasmids clusters associated with blood stream infections. Specifically, they should consider the findings of Reid et al., 2025 (BMC Genomics (2025) 26:57 <https://doi.org/10.1186/s12864-025-11226-4>) where an extensive F plasmid cluster analysis identified ColV and senB+ positive plasmids (Colla/archetype plasmids pUTI89, pRS218 & pEC14_114) in major ExPEC sequence types isolated from BSI. Carriage of these plasmids is reported ~ 60% of ExPEC recovered from BSI (Reid et al., 2025). It would also be important to see if the remaining F plasmids (non ColV/ Colla) recovered from BSI have interesting bacteriocin/microcin carriage.

The association of bacteriocins with ExPEC is intriguing and significant. ExPEC, unlike the other *E. coli* intestinal pathotypes are colonizing opportunistic pathogens. ExPEC colonizing factors include siderphores and other virulence associated genes and their carriage is considered a by product of commensalism (Le Gall et al., 2007. Extraintestinal virulence is a coincidental by-product of commensalism in B2 phylogenetic group *Escherichia coli* strains. *Mol Biol Evol* 24:2373–2384; Diard et al., 2010. Pathogenicity associated islands in extraintestinal pathogenic *Escherichia coli* are fitness elements involved in intestinal colonization. *J Bacteriol* 192:4885–4893). The authors could comment on how their data fits into this current view.

The association of bacteriocins with ExPEC is intriguing and significant. ExPEC, unlike the other *E. coli* intestinal pathotypes are colonizing opportunistic pathogens. ExPEC colonizing factors include siderphores and other virulence associated genes and their carriage is considered a by-product of commensalism (Le Gall et al., 2007. Extraintestinal virulence is a coincidental by-product of commensalism in B2 phylogenetic group *Escherichia coli* strains. *Mol Biol Evol* 24:2373–2384). The authors could comment on how their data fits into these current viewpoints which is reinforced in a recent paper by Condamine et al., *ISME J.* 2025 Jan 2;19(1):wrae245. doi: 10.1093/ismejo/wrae245. Lipworth, S. et al. The plasmidome associated with Gram-negative bloodstream infections: a large-scale observational study using complete plasmid assemblies. *Nat. Commun.* 15, 1612 (2024). Although this is a recently paper the authors also need to consider <https://doi.org/10.1038/s41467-025-57940-1>. This last reference pertains specifically to bacteriocins in a large plasmid cohort.

Version 1:

Reviewer comments:

Reviewer #1

(Remarks to the Author)
NA

Reviewer #2

(Remarks to the Author)

The reviewer thanks the authors for their detailed responses. Having addressed my previous comments (reviewer 2), the revised manuscript was carefully read, revealing a few additional minor comments and suggestions.

1. The authors added information (Fig S13, L80-85) about analyzed *E. coli* strains and bacteriocins; however, it is still not completely presented in the manuscript body. Figure S13 did not contain exact numbers of analyzed pathotypes and phylogroups. This has to be added in S13, but most importantly, also presented directly in the Methods/Results section (e.g., one sentence about % distribution of phylogroups).
2. Similarly, the exact set of analyzed bacteriocins has to be clearly defined (e.g., in Methods). The actual Tables 1 and S1 did not match completely. In Results (L105), any bacteriocins which were not found in the set should be mentioned.
3. There are also discrepancies in presented data. For example:
 - 2601 *E. coli* strains analyzed (L72), but the sum of *E. coli* with phylogroups is 1967 (L80-83).
 - 608 bacteriocinogenic strains were identified (L74), but 503 bacteriocinogenic strains are described with respect to phylogroups (L80-83).
4. The authors should be more exact in the manuscript, e.g., L91-92 (clarify "relatively low"), L196 (specify "majority"), L207 (rephrase "tended to encode"), L238 (show % of strains), and L417 (clarify "relatively rare").
5. L94-97: This statement (regarding the quality check for completeness of genomes) does not address the potential loss of bacteriocinogenic plasmids during DNA isolation (see previous review - Reviewer 2, point 4), which could be a reason for the low bacteriocinogenic prevalence observed in your study.
6. L393: *Yersinia* also produce bacteriocins, including pesticins, entericin, and colicin FY; the last is highly similar to colicin lb and is regulated by Fur.
7. In addition to bacteriocin sharing between *Escherichia-Salmonella* strains, this phenomenon is quite common for closely related *Escherichia-Shigella* strains. Some of the colicins are specifically active against pathogenic *Shigella*.

Reviewer #3

(Remarks to the Author)

I co-reviewed this manuscript with one of the reviewers who provided the listed reports. This is part of the Nature

Communications initiative to facilitate training in peer review and to provide appropriate recognition for Early Career Researchers who co-review manuscripts.

Reviewer #4

(Remarks to the Author)

This reviewer is satisfied that the authors have addressed the primary concerns of the original manuscript as specified in my review and indeed many of the other reviewers. The revised version is much improved and was a pleasure to read. I have no further comments

REVIEWER COMMENTS

Reviewer #1 (Remarks to the Author):

*This manuscript explores the role of bacteriocins in *E. coli*, especially their association with virulence and antimicrobial resistance (AMR) in an evolutionary perspective. Authors performed a comprehensive analysis of public genomic databases, combining phylogenetics, plasmidomics, and statistical modeling to propose that bacteriocin-encoding plasmids are key players in ExPEC pathogenesis and horizontal gene transfer. Most importantly, this work found that bacteriocin could be transferred along with resistance genes and virulence genes among pathogens. The narrative and data analysis are compelling and wonderful, but the analyses are correlative without experimental support. It's apparent this work focuses on data analysis, but such design should be discussed and the limitation should be stated clearly.*

Thank you for your positive and helpful feedback. We address each of your points below in turn.

Other specific minor comments were suggested as follows:

Specific comments

Line 78: What's the diversity of bacteriocin genes? Are they functioning in the same role or have different functions based on mutations? This background information should be clarified before genome analysis.

The bacteriocins of *E. coli* share similar domains but are diverse in that they target different outer membrane receptors and kill by delivering different cytotoxic domains to target cells, with a range of mechanisms of action. To help the reader distinguish between the different types, we have added a new table which describes the receptor targets and cytotoxic domains for each bacteriocin (Table 1). We have also added text (Line 79) explaining that plasmids can encode multiple different colicins and microcins.

Line 92-103: Authors should present the distribution of different bacteriocins in a better way. It's difficult to catch the main idea in this part. Do multiple weapons mean different bacteriocins or various mutants of the same weapon?

Multiple weapons means different bacteriocins that differ in their target and mechanisms of action. To make this clearer, we have additional text (line 79) and a table (Table 1), outlining what we mean by different bacteriocins and added additional text with an example of a colicin/microcin combination identified in the analysis (Line 119), to highlight that plasmids encode different colicins/microcins and not just mutants of the same bacteriocin/weapon.

Line 104-105: The study's findings do not support this conclusion.

We agree that we should not have stated "... these analyses suggest that there is a subset of *E. coli* strains that are particularly aggressive and specialised for bacterial warfare.". We have not shown that these strains are particularly aggressive compared to the other *E. coli* strains. Instead, the data only support the idea that these strains are in some way specialised for bacterial warfare. We have reworded the text to clarify this (Line 125-6) "...these analyses suggest that there is a subset of *E. coli* strains that are particularly specialised for bacterial warfare."

Lines 117–119, 482–490 The classification of strains into ExPEC, InPEC, and non-pathogenic is based on isolation site and the presence of a few virulence markers. However, hybrid strains and context-dependent virulence are common in E. coli. Justify the use of these categories and acknowledge their limitations. Consider complementing them with phylogroup or MLST-based clustering for more robust classification.

We agree that there are limitations with the predictions that one can make about *E. coli* pathotype based on genomics and metadata. Nevertheless, across such large datasets, these predictions remain a powerful tool for understanding *E. coli* and its pathogenesis. To make the pros and cons of our approach clearer, we have added the following text (Lines 141-146): “This classification is reliant upon the quality of the available metadata and does not distinguish between hybrid strains capable of causing multiple pathologies. Nevertheless, across the large dataset we have analysed, we see clear and interesting differences between the strain categories, which demonstrates the value in this approach. Moreover, our analysis agrees with known trends in *E. coli* phylogroups and sequence types.”

We have also compared our pathotype predictions to phylogroup and sequence-type predictions and highlighted where our pathotype predictions agree with the known phylogenetics e.g. the majority of our ExPEC strains where from phylogroup B2 and contained sequence types ST95 and ST131 which are known to cause a large amount of ExPEC diseases (Line 146-150). We have also added additional supplementary figures which describe the relationship between our pathotype predictions and phylogroups (Supplementary Figure 2)

Lines 146-150: “We find that phylogroup B2 is the most common group in ExPECs, E and B1 in InPECs and A in predicted gut symbionts which agrees with PCR and genomic analysis of *E. coli* pathotypes [56-58] (Supplementary Figure 2). Our methods also correctly classified common clinically important ExPEC lineages such as ST95, ST69 and ST131 and InPEC lineages such as ST11[59] and ST21[60].

Line 143: Rephrase this section title.

This sections title has been reworded (Line 174): “Bacteriocin plasmids are enriched in siderophores and other virulence factors”

Line 186: CTX-M-14 should not be in italic. Check gene names in the right format.

This has been corrected (Line 218).

Line 218: In this section, types of bacteriocin plasmids should be demonstrated in a logical way. What’s the most dominant AMR plasmid harboring bacteriocin? Such specific information is important than overall demonstration.

To aid comparison with previous literature we have added descriptions of different replicon types (and subtypes). This includes text describing the most common plasmid replicons where we find combinations of bacteriocin and antibiotic resistance genes (Line 219-222).

Line 219-222: “The combination of bacteriocins and AMR genes was not restricted to a single replicon but found on 42 different replicons (predicted by plasmid finder). The most common bacteriocin/antibiotic combination replicons predicted were Inc11-Iy, which encodes genes for Colla/Ib, followed by different IncF replicon subtypes.”

Lines 218–252: Claims about bacteriocin gene mobility and plasmid spread rely heavily on in silico tools (MOB-suite, PlasmidFinder), without experimental or historical data.

This is correct. However, MOB-Suite and Plasmidfinder are standard tools to predict plasmid mobility and are considered to be able do so with a high accuracy. We make it clear that our predictions are based on these tools and have included references to link our approach to other recent studies on plasmid mobility in large plasmid datasets and the impact of mobility of weaponry plasmids on bacterial competition.

Line 258: Authors can use different techniques to clarify plasmids, and the common one is to utilize replicon genes to screen. In E.coli, there are many well-known plasmids, such as typical AMR plasmids IncC/IncX3/IncQ/Incl2, etc. Did authors analyze plasmids based on that standard method? Also, the genetic environments of bacteriocin genes could be shown a little bit.

Bacteriocins were found on very diverse plasmids with each bacteriocin found on multiple replicon types, which limits the usefulness of using traditional replicon types in describing our results. However, to bring our results into line with other studies, we have performed new analysis to show how our findings relate to replicon typing. Additional text describing bacteriocin plasmids replicons can be found throughout the text (Lines: 219-222, 258-264)

Line 346: In this part, I do not see the common plasmids in Enterobacteriaceae. What's the reason?

In this section we analysed 11274 plasmids. To represent all of these plasmids within a single figure unfortunately would make the figure unreadable. We therefore only show plasmid clusters which contain at least one colicin-like plasmid, which means that some common plasmids are not shown. We now make this clear as follows:

Line 1104-1105: "For clarity, only plasmid clusters with contain at least 1 colicin plasmid are shown."

Line 374: Collb and Collb plasmids? Furthermore, the conclusion stated in this sentence is not supported by the current data presented in Figure 5b.

Thank you. This has been corrected to Colla and Collb, and the text has been revised as follow to make it more clearly fit the data in Figure 5b:

Lines 423-426: "By contrast, edges between *Escherichia* and *Salmonella* colicinogenic plasmids are common, typically involving Colla and Collb plasmids, and comprised 80% of all edges between species in our analysis (Figure 5b and Supplementary Figure 11)."

We have also added an additional supplementary figure (Supplementary Figure 11), which visualises the proportion of edges shared between genera with the associated numbers in order to allow the reader to more easily see which species share the most edges.

Line 386: It seems that authors did not analyze chromosomal bacteriocin compared to plasmids.

The reviewer is correct that our analysis here focusses on bacteriocin plasmids and we have altered the text to make this clear (Line 436):

Line 436: “The bacteriocins of *E. coli* occur both on the chromosome and on plasmids, and we find evidence that the plasmid-encoded bacteriocins can readily move between both strains and plasmids.”

Line 416: This research is a good analytical work, but experiments were not performed to support such results. This limitation could be presented as an important section to imply a few future directions.

We have now highlighted the strengths of our analysis and discussed future avenues for testing in the lab as requested (Line:450-455).

Line 450-455: “More generally, our findings suggest that iron-targeting bacteriocins, siderophores and other virulence factors can work in concert to help pathogens to both colonise diverse communities, such as the human microbiome, and cause disease. Experimental tests are needed to confirm this hypothesis along with the potential for siderophores and iron-targeting bacteriocins in particular to work in synergy to enable persistence of ExPEC strains in the microbiome [98-100].”

Figure 1B The scale bar is missing from the phylogenetic tree.

The scale bar has been returned to this figure.

Several figures or figure legends are overly simplistic, lacking necessary descriptions and statistical analyses of differences. Please review and revise accordingly.

Figure legends have been expanded to give better detail for Figures 1, 2, 4, and 5. This revision includes discussing statistical analysis of differences. Figures 1, 2,3, and 4 have been altered to ensure the correct and consistent gene names for bacteriocins.

Some references have formatting inconsistencies and are missing page numbers. Please check and revise accordingly.

Reference have been reformatted for consistency.

From Ruichao Li

Reviewer #2 (Remarks to the Author):

In the manuscript “Bacterial warfare is associated with virulence and antimicrobial resistance,” the authors analyzed prevalence of bacteriocins across E. coli genomes available in databases using bioinformatic tools. E. coli is a very heterogeneous species, and its evolution and ecology are not fully understood. Therefore, complementary analyses at the population level are crucial. However, I have concerns about the relevance and presentation of their results.

Thank you for your helpful and constructive comments. We go through each in turn.

Major comments:

1. The authors analyzed set of E. coli genomes with well-defined inclusion/exclusion criteria. However, there are missing epidemiological information about used strains. Particularly, prevalence of individual phylogroups, serotypes, and pathotypes has to be added. For example, how many ExPEC strains belonging to phylogroup B2 were included in the analyzed set. This characterization is important for interpretation of the findings (see below).

To better describe the dataset, we have added 2 additional supplementary figures. Supplementary Figure 2 visualises the relationship between our pathotype predictions and phylogroups. Supplementary Figure 14 outlines the distribution of different sequence types in our dataset. We have also added additional text describing the relationship between our pathotypes predictions and phylogroup and sequence types.

Lines 145-150: “...our analysis agrees with known trends in *E. coli* phylogroups and sequence types. We find that phylogroup B2 is the most common group in ExPECs, E and B1 in InPECs and A in predicted gut symbionts which agrees with PCR and genomic analysis of *E. coli* pathotypes [56-58] (Supplementary Figure 2). Our methods also correctly classified common clinically important ExPEC lineages such as ST95, ST69 and ST131 and InPEC lineages such as ST11[59] and ST21[60].”

2. Similarly, the authors did not include a list of the analyzed bacteriocins in the methods section. The different bacteriocin subsets are presented across the figures. And the list of bacteriocins appears to be incomplete (e.g., bacteriocins A, E4, E5, E9, U, R, and microcin PDI are missing from Figure 1). Furthermore, there are errors in classifications – e.g., MccB should be MccB17, and Col J should be ColJs. The list of known bacteriocins is available in reviews (e.g., doi: 10.1080/14787210.2020.1816824, doi: 10.3389/fmicb.2020.586433).

Certain bacteriocins are extremely rare in *E. coli* (or in the case of ColA, not actually found in *E. coli* but found in closely related species), and though we included them in our initial search, we were unable to find them in our datasets, including the bacteriocins mentioned (Colicins A, E4, E5, E9, and colicins U/R which are identical). We now state these considerations explicitly:

Lines 538-539: “Some colicins are very rare and though we they were included in our initial search, they were not identified even in our dataset of 2601 genomes.”

Microcin PDI was not included in our analysis as it requires proximity between attacker and prey, (proximity dependent inhibition) suggesting that it does not function as a typical bacteriocin.

A full list of the bacteriocins searched for is provided in Supplementary Table 1. Nomenclature for ColJs and MccB17 has been corrected throughout.

3. Furthermore, the authors analyzed over 5,000 plasmids from a separate database. How

were these plasmids assigned to specific E. coli genomes, and were unassigned plasmids omitted from the study? How many E. coli genomes did not contain plasmid(s)?

The >5000 plasmids are from PLSDDB a database of sequences plasmids. They are not assigned to an individual genome, but the species of isolation is captured from metadata (<https://doi.org/10.1093/nar/gky1050>). As some plasmids are mobile between species, we examined all plasmids from *Klebsiella*, *Salmonella*, *Escherichia*, *Shigella* and *Citrobacter* to investigate colicin movement between species. Additional text has been added to better describe the plasmid database (Lines 562-565). Note that in another analysis in our study (Figs 1 and 2), we use a different database where plasmids and chromosomes were sequenced together. This dataset does allow one to analyse entire genomes of bacteriocinogenic strains to look for associations with the chromosome and different pathotypes. In this dataset we found 29.4% of strains did not contain a plasmid, which is similar to other experimental studies (DOI:10.1111/1462-2920.13552: 26% of *E. coli* strains had no plasmid).

4. The reviewer's previous comments relate to the finding that bacteriocin production was observed in only 20% of the E. coli strains analyzed. This contrasts with numerous prior studies across diverse E. coli sets, which reported significantly higher levels of bacteriocinogeny (approximately 50%). Notably, the prevalence of bacteriocin genes reached up to 75% in ExPEC isolates. There are two potential explanations for this discrepancy. First, there may be a sampling bias, with the majority of analyzed strains originating from environmental conditions and laboratory and representing non-pathogenic isolates, which typically belong to phylogroups A/B1 and have previously been associated with lower bacteriocin levels. Another possibility is that some analyzed genomes was compromised by plasmid loss during DNA isolation, a common issue in older sequencing data, which likely comprises part of this study.

Estimates of bacteriocinogenicity in *E. coli* vary greatly between studies depending on sampling and identification method. Many previous studies have focus solely on a single environment or pathotype of *E. coli*. Our analysis with a very large and phylogenetically broad *E. coli* dataset shows that bacteriocinogenicity varies greatly between phylogroup, pathotype and sequence type. We have added additional text that puts our estimates of bacteriocin carriage into context:

Line 87-98: "Estimations of bacteriocin carriage rates in the *E. coli* literature vary greatly from 15%-60%. However, sampling and method of identification also vary between estimates, including several that are based exclusively on human fecal samples or specific *E. coli* pathotypes [33, 34]. Our analysis of thousands of *E. coli* genomes and plasmids suggests that large differences in bacteriocin abundance among phylogroups may explain the different estimates of bacteriocin carriage among studies, including our own. For example, the high prevalence of phylogroup A in our dataset, which has relatively low bacteriocin carriage may help to explain why our estimate is relatively low as compared to studies focussed on other phylogroups and pathogenic *E. coli*[34-37]. Another possible factor is that plasmid sequences may have been lost from certain samples, although our quality control methods that selects for complete genomes should mean this is relatively unlikely. Whatever the case, our estimates of bacteriocin carriage are likely to be conservative."

5. Another major issue is the limited novelty of this study, as numerous previous studies have experimentally demonstrated that E. coli bacteriocins are: i) encoded on plasmids and horizontally transferred among E. coli strains; ii) encoded on plasmids that frequently harbor

diverse virulence factors, including adhesins, invasins, and iron-acquisition systems; iii) regulated via SOS-damage and iron limitation; and iv) associated with pathogenic E. coli, particularly strains belonging to the B2 phylogroup and responsible for extraintestinal infections in humans.

Our study does indeed fit well with previous experimental work. However, our approach allows us to study vast numbers of bacteriocins and genomes in a manner that is not possible via experiments. From this, we are able to not only confirm outcomes from previous experimental studies, but also document broad patterns in bacteriocin ecology and evolution, and the links to pathogenicity. In particular, we show that there are statistical association between pathogenicity type and the use of certain bacteriocins, demonstrating the link between bacteriocin carriage and disease. By analysing the distribution of bacteriocin orphan immunity proteins and toxin-antitoxin systems, we find clear support for the importance of these bacteriocins for interbacterial warfare. We also document the frequent movement of bacteriocin plasmids between *E. coli* and Salmonella, further cementing the link to virulence.

6. The authors performed a considerable set of bioinformatic analyses; however, most of the presented data lacks informativeness. For example, the authors present a scheme of plasmid clusters, but the exact names for individual clusters are missing.

To increase informativeness, we have now added additional analysis to put our results in the context of Inc replicons and we have added additional references to describe our results in the context of previous literature.

The Louvain clustering algorithm used in our analysis is excellent for visualising plasmid diversity, and seeing the evidence for the association of siderophores with bacteriocins and the horizontal transfer of bacteriocins between diverse plasmids. However, we have decided to refrain from using the names of individual clusters in the main text as the clusters in this method are fluid in the sense that re-performing the analysis would name clusters differently. However, to help the reader learn more about the clusters, we have added an additional supplementary figure with all plasmids clusters labelled (Supplementary Figure 6) and provided the details for each of the 5373 individual plasmids in the analysis (Supplementary table 4). This approach is typical for Louvain clustering work (10.7554/eLife.85302) to help the reader interpret the analysis.

Reviewer #3 (Remarks to the Author):

Reviewer #4 (Remarks to the Author):

This is an interesting and well analysed body of work that speaks to the functions of bacteriocins and their role in E. coli ecology. In the first instance it is great to see novel work being undertaken to study E. coli ecology as it is an area that has been somewhat neglected. The overarching conclusions is that bacteriocins are important for bacterial ecology, competition and virulence. The manuscript is well written and there are no typographic errors.

The major findings of this work are that: i) bacteriocins are mobile being facilitated in their movement by the actions of flanking IS elements (compound transposons) and on plasmids; ii) E. coli bacteriocins are abundant in pathogenic strains and linked to known virulence factors such as siderophores, adhesins and enterotoxins; and iii) plasmids containing orphan immunity proteins, which protect against bacteriocin attack without producing toxins, supporting the importance of bacteriocins for bacterial competition.

The authors conclude their abstract with the following: Our work suggests that the bacteriocins of E. coli are important antibacterial weapons for dangerous antimicrobial-resistant strains.

Thank you for the positive assessment of our study. We discuss each of your comments below and respond.

Major concerns:

The major weakness of this manuscript is that the authors could do a much better job relating their findings to current thinking regarding the ecology of E. coli particularly ExPEC as well as the role of bacteriocins (if any) among well described ExPEC plasmids in ExPEC disease. Moreover the authors should consider a case study in F plasmids (known for carriage of VAGs), adding F plasmid MLST replicon types, and compare their finding and how they impact existing and recent F plasmid clustering studies.

In this regard the authors could add a Table of the major E. coli Sequence types. This refers to the following statement (line79): Bacteriocins are also very widely distributed across E. coli strains, being present in all E. coli phylogroups (G (75%, 21/28), B2 (46.3%, 173/374) and B1 (34.4%, 166/482)), and >90% (38/44) of all sequence types (STs) which had more than 10 representatives (Figure 1a). The readership would like to know more about these (major) sequence types and Fig 1 or the manuscript more broadly does not provide any information about this.

Thank you for this suggestion. To better convey the diversity in our dataset and link to known sequence types, we have added additional figures which highlight the relationship between our pathotype predictions and phylogroups (Supplementary Figure 2) and the distribution of sequence types (Supplementary Figure 14). Individual sequence type predictions are also added to Supplementary Table 1.

Furthermore, the authors should consider relating their finding to the major ExPEC F plasmids clusters associated with blood stream infections. Specifically, they should consider the findings of Reid et al., 2025 (BMC Genomics (2025) 26:57 <https://doi.org/10.1186/s12864-025-11226-4>) where an extensive F plasmid cluster analysis identified ColV and senB+ positive plasmids (Colla/archetype plasmids pUTI89, pRS218 & pEC14_114) in major ExPEC sequence types isolated from BSI. Carriage of these plasmids is reported ~ 60% of ExPEC recovered from BSI (Reid et al., 2025). It would also be important to see if the remaining F plasmids (non ColV/ Colla) recovered from BSI have interesting bacteriocin/microcin carriage.

Though ExPEC strains were not the primary focus of our study, we agree that it would be beneficial to the reader to link our results up better to the ExPEC literature. To this end, we added new analyses of our pColV plasmids based upon IncF replicon sequence types (IncF RSTs, which were extensively used in Reid et. al. 2025) and describe the diversity we observe

in our dataset in comparison. We find a similar level of MccV carriage in our ExPEC strains as the Reid study (Our analysis: 15.8% vs 19.6% in the Reid Study). We also have added new text to highlight how our analyses fit with previous studies:

Lines 253-264: “Our analyses fit with previous work on *E. coli* bacteriocins. For example, plasmids encoding the microcin MccV (known as pColV plasmids) have been previously identified in ExPEC strains and found to be maintained in lineages for long periods [80, 81]. We also found pColV plasmids to be common in ExPEC strains, with 15.8% of predicted ExPEC strains encoding a MccV+ plasmid [81]. Previous research also has shown that ExPEC strains commonly possess plasmids with specific IncF replicons [80-82]. Analysing our ExPEC strains with pColV plasmids, we find the commonly reported IncF replicon sequence types including F18:A-B-, F18:A-B1, F24:A-B1, F24:A-B-, and F2:A-B1. In addition, we found pColV plasmids commonly encode multiple bacteriocins, with 58% of pColV plasmids possessing additional bacteriocins such as ColM, Colla, and Collb. Moreover, other ExPEC strains in the dataset encoded Colla, Collb and ColM bacteriocins on a range of MccV- IncF and IncI plasmids.”

The association of bacteriocins with ExPEC is intriguing and significant. ExPEC, unlike the other E. coli intestinal pathotypes are colonizing opportunistic pathogens. ExPEC colonizing factors include siderophores and other virulence associated genes and their carriage is considered a by-product of commensalism (Le Gall et al., 2007. Extraintestinal virulence is a coincidental by-product of commensalism in B2 phylogenetic group Escherichia coli strains. Mol Biol Evol 24:2373–2384; Diard et al., 2010. Pathogenicity associated islands in extraintestinal pathogenic Escherichia coli are fitness elements involved in intestinal colonization. J Bacteriol 192:4885–4893). The authors could comment on how their data fits into this current view.

We have expanded our discussion of the importance of gut colonization in our conclusion, and highlighted the possible role of siderophores in maintaining strains in the gut and the influence this could have on disease. We have also added additional references, which are consistent with these ideas:

Lines 446-450: “Under these conditions, bacteria upregulate the expression of iron uptake receptors, which increases their susceptibility to iron-uptake targeting weaponry [20]. The use of such weaponry, therefore, may give strains a fitness advantage under low iron conditions [20], such as those found in the inflamed gut. From the gut, strains can then spread to extra-intestinal infection sites such as the urinary tract [95-97].”

Lines 456-461: “However, consistent with these ideas, gut colonization is an important step for the pathogenesis of extraintestinal *E. coli* in animal studies and extra-intestinal virulence factors can increase survival in the gut [77, 78, 98-103]. Disruption of gut colonization represents a potential therapeutic strategy to prevent extra-intestinal *E. coli* infections [104]. Our work suggests that understanding the links between bacterial warfare and virulence will aid in this endeavour.”

The association of bacteriocins with ExPEC is intriguing and significant. ExPEC, unlike the other E. coli intestinal pathotypes are colonizing opportunistic pathogens. ExPEC colonizing factors include siderophores and other virulence associated genes and their carriage is considered a by-product of commensalism (Le Gall et al., 2007. Extraintestinal virulence is a coincidental by-product of commensalism in B2 phylogenetic group Escherichia coli strains. Mol Biol Evol 24:2373–2384). The authors could comment on how their data fits into these

current viewpoints which is reinforced in a recent paper by Condamine et al., ISME J. 2025 Jan 2;19(1):wrae245. doi: 10.1093/ismejo/wrae245. Lipworth, S. et al. The plasmidome associated with Gram-negative bloodstream infections: a large-scale observational study using complete plasmid assemblies. Nat. Commun. 15, 1612 (2024). Although this is a recently paper the authors also need to consider <https://doi.org/10.1038/s41467-025-57940-1>. This last reference pertains specifically to bacteriocins in a large plasmid cohort.

We thank the reviewer for highlighting recent literature. We have included additional text to describe our results in context of this literature and included these references to support our analysis throughout the text e.g. Lines 253-264 and Lines 448-458.

Lines 253-264: “Our analyses fit with previous work on *E. coli* bacteriocins. For example, plasmids encoding the microcin MccV (known as pColV plasmids) have been previously identified in ExPEC strains and found to be maintained in lineages for long periods [80, 81]. We also found pColV plasmids to be common in ExPEC strains, with 15.8% of predicted ExPEC strains encoding a MccV+ plasmid [81]. Previous research also has shown that ExPEC strains commonly possess plasmids with specific IncF replicons [80-82]. Analysing our ExPEC strains with pColV plasmids, we find the commonly reported IncF replicon sequence types including F18:A-B-, F18:A-B1, F24:A-B1, F24:A-B-, and F2:A-B1. In addition, we found pColV plasmids commonly encode multiple bacteriocins, with 58% of pColV plasmids possessing additional bacteriocins such as ColM, Colla, and Collb. Moreover, other ExPEC strains in the dataset encoded Colla, Collb and ColM bacteriocins on a range of MccV- IncF and IncI plasmids.”

Lines 447-461: “The use of such weaponry, therefore, may give strains a fitness advantage under low iron conditions [20], such as those found in the inflamed gut. From the gut, strains can then spread to extra-intestinal infection sites such as the urinary tract [95-97]. More generally, our findings suggest that iron-targeting bacteriocins, siderophores and other virulence factors can work in concert to help pathogens to both colonise diverse communities, such as the human microbiome, and causes disease. Experimental tests are needed to confirm this hypothesis along with the potential for siderophores and iron-targeting bacteriocins in particular to work in synergy to enable persistence of ExPEC strains in the microbiome [98-100].

However, consistent with these ideas, gut colonization is an important step for the pathogenesis of extraintestinal *E. coli* in animal studies and extra-intestinal virulence factors can increase survival in the gut [77, 78, 98-103]. Disruption of gut colonization represents a potential therapeutic strategy to prevent extra-intestinal *E. coli* infections [104]. Our work suggests that understanding the links between bacterial warfare and virulence will aid in this endeavour.”

REVIEWERS' COMMENTS

Reviewer #1 (Remarks to the Author):

NA

Reviewer #2 (Remarks to the Author):

The reviewer thanks the authors for their detailed responses. Having addressed my previous comments (reviewer 2), the revised manuscript was carefully read, revealing a few additional minor comments and suggestions.

1. The authors added information (Fig S13, L80-85) about analyzed E. coli strains and bacteriocins; however, it is still not completely presented in the manuscript body. Figure S13 did not contain exact numbers of analyzed pathotypes and phylogroups. This has to be added in S13, but most importantly, also presented directly in the Methods/Results section (e.g., one sentence about % distribution of phylogroups).

We have added exact numbers to the text in both the results and methods and they can also be found in the source data. Exact numbers can be found on lines 83-86 and lines 499-501:

Lines 83-86: “Bacteriocins are also very widely distributed across E. coli strains and are present in all E. coli phylogroups: 75% of phylogroup G plasmids carried bacteriocins, (21/28), 46.3%, of B2 (173/374), 34.4 of B1 (166/482), 24% of C (22/93) and 12% of A (121/990) (Other or un typeable phylogroups: 105/634).”

Lines 499-501: “Genomes covered all known E. coli phylogroups (A: 990, B1: 482, B2: 374, C: 93, D: 234, E: 301, E/I: 1, F: 84, G:28, I: 4, Unknown: 10).”

2. Similarly, the exact set of analyzed bacteriocins has to be clearly defined (e.g., in Methods). The actual Tables 1 and S1 did not match completely. In Results (L105), any bacteriocins which were not found in the set should be mentioned.

Bacteriocins which were included in the search but were not identified have been listed in the methods section, Lines 515-516:

Lines 515-516: “ Some colicins are very rare and though we they were included in our initial search, they were not identified even in our dataset of 2601 genomes (e.g. ColA, ColE4 and ColR, ColZ, ColE9)”

3. *There are also discrepancies in presented data. For example:*

- *2601 E. coli strains analyzed (L72), but the sum of E. coli with phylogroups is 1967 (L80-83).*
- *608 bacteriocinogenic strains were identified (L74), but 503 bacteriocinogenic strains are described with respect to phylogroups (L80-83).*

Lines 80-83 (now lines 83-86) is not an exhaustive list of all phylogroups as it would interrupt the text. To solve this discrepancy we have added a category of ‘Other’ to make the numbers add correctly. A full list of phylogroups can also be found in the methods section (Lines 499-501), Supplementary Data and Source Data.

4. *The authors should be more exact in the manuscript, e.g., L91-92 (clarify "relatively low"), L196 (specify "majority"), L207 (rephrase "tended to encode"), L238 (show % of strains), and L417 (clarify "relatively rare").*

The manuscript has been altered to be more specific mentioning percentages, exact numbers, or references to the figure with the data where possible. Lines 91-92 includes the percentage of phylogroup A bacteriocin carriage. Line 196 includes a reference to Supplementary Figure 3 and Line 417 now references Supplementary Figure 11, which shows the proportion of edges between species.

5. *L94-97: This statement (regarding the quality check for completeness of genomes) does not address the potential loss of bacteriocinogenic plasmids during DNA isolation (see previous review - Reviewer 2, point 4), which could be a reason for the low bacteriocinogenic prevalence observed in your study.*

We have text which discusses the possibility of plasmid DNA loss during isolation (Lines 100-101).

6. *L393: Yersinia also produce bacteriocins, including pesticins, entericin, and colicin FY; the last is highly similar to colicin Ib and is regulated by Fur.*

The reviewer is correct that Yersinia produce colicin-like bacteriocins as well as Pseudomonads and other Families within Gammaproteobacteria, however they are not as well

studied as the colicin-like bacteriocins from Enterobacteriaceae. We include references which mention these other bacteriocins.

7. In addition to bacteriocin sharing between Escherichia-Salmonella strains, this phenomenon is quite common for closely related Escherichia-Shigella strains. Some of the colicins are specifically active against pathogenic Shigella.

The proportion of edges shared between each genera is shown in Supplementary Figure 11.

Reviewer #3 (Remarks to the Author):

Reviewer #4 (Remarks to the Author):

This reviewer is satisfied that the authors have addressed the primary concerns of the original manuscript as specified in my review and indeed many of the other reviewers. The revised version is much improved and was a pleasure to read. I have no further comments